

# Fault slip potential induced by fluid injection in the Matouying EGS field, Tangshan seismic region, North China

**Chengjun Feng[1,2], Guangliang Gao[3,4], Shihuai Zhang[5], Dongsheng Sun[1,2], Siyu Zhu[1], Chengxuan Tan[1,2], Xiaodong Ma[5]**

[1]Institute of Geomechanics, Chinese Academy of Geological Sciences, Beijing, 100081, China

[2]Key Laboratory of Active Tectonics and Geological Safety, Ministry of Natural Resources, Beijing, 100081, China

[3]China University of Petroleum, Qingdao, 266580, China

[4]Jidong Oilfield Company, PetroChina, Tangshan, 063000, China

[5]Department of Earth Sciences, ETH Zürich, Zürich, 8092, Switzerland

**Correspondence**: Chengjun Feng (feng2010618@aliyun.com)

**Abstract.** The Tangshan region is one of the most seismically active areas in the North China, and the 1976 $M$ 7.8 earthquake occurred on July 28th near the Tangshan fault zone. The Matouying Enhanced Geothermal Systems (EGS) field is located ~90 km away from Tangshan City. Since the late 2020, preliminary hydraulic stimulation tests have been conducted at depths of ~3965–4000 m. Fluid injection into geothermal reservoir facilitates heat exchanger system. However, fluid injection may also induce earthquakes. In anticipation of the EGS operation at the Matouying uplift, it is essential to assess how the fault slip potential of the nearby active and quiescent faults will change in the presence of fluid injection. In this study, we first characterize the ambient stress field in the Tangshan region by performing stress tensor inversions using 98 focal mechanism data ($M_L \geq 2.5$). Then, we estimate the principal stress magnitudes near the Matouying EGS field by analyzing in situ stress measurements at shallow depths (~600–1000 m). According to these data, we perform a quantitative risk assessment using the Mohr-Coulomb framework in order to evaluate how the main active faults might respond to hypothetical injected-related pore pressure increases due to the upcoming EGS production. Our results mainly show that most earthquakes in the Tangshan seismic region have occurred on the faults that have relatively high fault slip potential in the present ambient stress field. At well distances of less than 15 km, the probabilistic fault slip potential on most of the boundary faults increase with continuing fluid injection over time, especially on these faults with well distances of ~6–10 km. The probabilistic fault slip potential increases linearly with the fluid injection rate. However, the FSP values decrease exponentially with increased unit permeability. The case study of the Matouying EGS field has important implications for the deep geothermal exploitation in China, especially for Gonghe EGS (in Qinghai province) and Xiong'an New Area (in Hebei province) geothermal reservoirs that are close to


the Quaternary active faults. Ongoing injection operations in the regions should be conducted with these understandings in mind.

**1 Introduction**

Enhanced geothermal systems (EGS) are a promising source of renewable energy for a decarbonizing world and can provide a valuable contribution to the production of renewable energy (Lee et al., 2019). The EGS technologies exploit geothermal resources through hydraulic stimulation, which involves the injection of high-pressure cold water into the target formation in order to increase the unit permeability by creating new fractures or causing preexisting fractures to widen (Terakawa et al., 2012; Grigoli et al., 2018; Lee et al., 2019). To economically produce electricity and heat with an EGS, it is necessary to employ an efficient hydraulic subsurface heat exchanger system that can circulate through the hot rock that hosts the permeable fracture network (Bromley et al., 1987; Häring et al., 2008).

The industrial process of hydraulic stimulation involves creating tensile fractures and subsequently increasing the permeability of the target rock formations via the controlled injection of pressurized fluid (Ellsworth, 2013). However, while the injection of fluid into reservoir rocks facilitates oil and gas recovery, plays a key role in EGS, and aids in the disposal of wastewater and $CO_2$ gas, fluid injection may also induce earthquakes (Shapiro et al., 2005; Evans et al., 2012; Zoback and Gorelick, 2012; Ellsworth, 2013; Zang et al., 2014; McGarr et al., 2015; Walsh and Zoback, 2015; Lei et al., 2017; Kim et al., 2018; Lee et al., 2019). Seismic events caused by fluid injection are a possible hazard faced by nearly all engineering endeavors that result in changes to the ambient subsurface stress or pore pressure (Evans et al., 2012).

For the past 40 years, induced seismicity has been documented in geothermal settings such as the Philippines (Bromley et al., 1987), Japan (Nagana et al., 1994), Kenya (Simiyu, 1999), North and South America (Henderson et al., 2002), Australia (Baisch et al., 2006), and New Zealand (Hunt and Latter, 1982). Evans et al. (2012) compiled a survey of induced seismic events caused by fluid injection in European geothermal reservoirs. Annually, thousands of seismic events (with local magnitudes $M_L <$ 2.0) are generated during the exploitation of geothermal fields (Evans et al., 2012). Furthermore, EGS case studies have demonstrated that injecting water into basement rock mass may also produce large earthquakes (with moment magnitude $M \geq 3.0$). For example, in 2006 and 2007, four $M$ 3.0 earthquakes were caused by the high-pressure injection of water into impermeable basement rocks beneath Basel, Switzerland (Deichmann and Giardini, 2009; Terakawa et al., 2012; Ellsworth, 2013; McGarr et al., 2015). In 2017, a $M$ 5.5 earthquake occurred near an EGS drill site in Pohang, South Korea (Zang et al., 2014; Kim et al., 2018; Grigoli et al., 2018; Lee et al., 2019; Woo et al., 2019). Geological and geophysical data from this study area suggests that the Pohang earthquake was caused by the injection of fluid directly into the near-critically stressed Yangsan fault zone (Kim et al., 2018; Grigoli et al., 2018).

On June 30th, 2019, the No. 2 Exploration Team of the Hebei Bureau of Coal Geological Exploration in China announced that their team had drilled to a depth of 3965 m into the Matouying (MTY) uplift (Fig. 1). With a temperature of 150°C, this area of hot dry rock is located ~90 km away from Tangshan City in northern China (Qi et al., 2020; Zhang et al., 2020). At the time, geological



prospecting surveys indicated that the two target areas for the MTY EGS field had areas of 80 km$^2$ (at a
depth of 4000 m) and 500 km$^2$ (at a depth of 5000 m) and may yield as much as equivalent of ~2.8
billion and ~22.8 billion tons of standard coal, respectively. In 2020, preliminary hydraulic stimulation
tests were conducted at depths of ~3965–4000 m (Qi et al., 2020).

5       The Tangshan region is one of the most seismically active areas in the North China basin (Jiang,

2006; Feng et al., 2019). On July 28th, 1976, a $M$ 7.8 earthquake (focal depth of 10 km) struck the city
of Tangshan in northern China, 160 km east of Beijing City (Fig.1) (Guo et al., 1977; Chen et al., 1979).
The largest aftershock (the $M$ 7.1 Luanxian earthquake) occurred on the same day approximately 45 km
northeast of the mainshock location (Fig. 1). Another large aftershock (the $M$ 6.9 Ninghe earthquake)
near Ninghe County occurred on November 15th, 1976 (Fig. 1) (Nábělek et al., 1987; Huang and
Yeong, 1997). The Tangshan $M$ 7.8 earthquake, which was one of the most devastating earthquakes in
the last 100 years worldwide, was responsible for more than 242,000 deaths and 800,000 injuries, and
ultimately destroyed the city of Tangshan (Li et al., 2018). Northern China is marked by unusually
active intraplate seismicity and Quaternary tectonic movement (Ye et al., 1985; Nábělek et al., 1987).
Previous studies indicate that most faults in the Tangshan seismic region, such as the Tangshan fault
belt, the Luanxian-Laoting fault, the Changli-Ninghe fault, and the Jiyunhe fault, were created by late
Pleistocene-Holocene tectonic activity (Li et al., 1998; You et al., 2002; Jiang, 2006; Guo and Zhao,
2019). While the seismic activity in the Tangshan region has decreased over the past 40 years (Zhang et
al., 2017), some $M_L$ 4.0–5.0 earthquakes have still occurred in the past 10 years (e.g., 5/28/2012 ($M_L$
5.2), 9/14/2015 ($M_L$ 4.2), 9/10/2016 ($M_L$ 4.2), 3/24/2017 ($M_L$ 4.4), 8/2/2019 ($M_L$ 4.4)) (Yang et al., 2016;
Lin et al., 2017; Feng et al., 2019; Fan et al., 2019).
In anticipation of the EGS operation at the MTY uplift in the Tangshan seismic region, it is
essential to assess how the fault slip potential of the nearby active and quiescent faults will change in
the presence of fluid injection. In this study, we first characterize the ambient stress field in the
Tangshan region by performing stress tensor inversions using focal mechanism data ($M_L \geq 2.5$) from the
past 14 years. With these inversions, we determine the principal compressive stress orientations, the
prevailing stress regime, and the critical coefficients of friction throughout our study area. By analyzing
in situ stress measurements at shallow depths (~600–1000 m) in the Tangshan seismic region (Tan et al.,
2014; Tan et al., 2015; Niu et al., 2015; Feng et al., 2019), we estimate the principal stress magnitudes
near the MTY EGS field. We then perform a quantitative risk assessment using the Mohr-Coulomb
framework in order to evaluate how the main active faults in the Tangshan seismic region might
respond to hypothetical injected-related pore pressure increases due to the upcoming MTY EGS
production. This assessment is based on the FSP v.1.0 software package from Stanford Center for
Induced and Triggered Seismicity of Stanford University (Walsh and Zoback, 2015; Walsh and Zoback,
2016; Lund Snee and Zoback, 2018). In our analysis, we use only publicly available information
related to the most active faults in the Tangshan seismic region. Finally, we conduct a seismic hazard
assessment by predicting the maximum moment magnitudes of injection-induced seismic events in the
MTY EGS field in response to different net fluid injection volumes.
**2 Tectonics and seismicity in the Tangshan seismic region**





The Tangshan seismic region is situated in the northern part of the North China Plain. This large basin
began to form in the early to middle Eocene (Ye et al., 1985). As with many other basins, rifting was
the primary mode of tectonic activity in the initial stages of basin development (Shedlock et al., 1987).
This basin is bounded to the north by Yanshan Mountains and to the east and the south by Bohai Bay.
The topography is higher in the northern part of the basin than it is in the southern part of the basin
(Guo and Zhao, 2019) (Fig. 1). Structurally, the Tangshan seismic region is a part of the Kailuan
sag-fold system in the Yanshan fold belt; the basement material is the NE-trending Kaiping
synclinorium, a Yanshan stage formation that consists of Paleozoic rocks (Guo et al., 2011).
Geophysical prospecting (Hao et al., 1998; Li et al., 1998; Li et al., 2009; Yang et al., 2010; Liu et al.,
2011; Ran et al., 2013; Zhang et al., 2013), geological mapping (Zheng et al., 1981; Gao et al., 2001;
Guo et al., 2011; Guo and Zhao, 2019), and identification of geomorphic features (Qiu et al., 2005)
have revealed the existence of various fault systems in this region (Fig. 1); the orientations of these
fault systems are NEN (e.g., the Lulong fault - $F_6$), ENE (e.g., the Tangshan fault belt - $F_4$, the Ye'jituo
fault - $F_3$, the Changli-Ninghe fault - $F_5$, the Cangdong fault - $F_{14}$, and the Xi'nanzhuang fault - $F_{11}$),
NW (e.g., the Ji'yunhe fault - $F_2$, the Luanxian-Laoting fault - $F_7$, and the Bai'gezhuang fault - $F_8$),
WNW (e.g., the Lengkou fault - $F_{17}$) and nearly EW (e.g., the Han'gu fault - $F_{13}$ and the Haihe fault -
$F_{12}$). The NE-trending faults, which are the most prominent faults in the area, run throughout the entire
length of the Tangshan seismic region (Ye et al., 1985). Some of the faults sets ($F_2$, $F_3$, $F_4$, $F_5$, and $F_7$)
divide the basin into rhombic and triangular blocks. Furthermore, these faults show signs of recent
normal and strike-slip movement (Nábělek et al., 1987; Liu et al., 2011; Feng et al., 2019).
Based on the co-seismic crustal deformation, previous studies have concluded that the mainshock
of the 1976 $M$ 7.8 Tangshan earthquake sequence ruptured along a right-lateral fault with a strike of
N30°–53°E and a dip of 76°–89° SE (Butler et al., 1979; Wan et al., 2017). The Luanxian $M$ 7.1
earthquake was associated with pure normal faulting on a plane with a strike of N30°W and a dip of
45°–53° NE (Huang and Yeong, 1997; Wan et al., 2017). The Ninghe $M$ 6.9 earthquake was
characterized by left-lateral strike-slip faulting with a small normal component on a NW-striking
(N36°W) fault plane with a dip of 67° NE (Huang and Yeong, 1997; Wan et al., 2008; Wan et al.,
28  2017).

**3 The Coulomb failure criterion**
The Mohr-Coulomb failure criteria is a useful framework for understanding how increasing the pore
fluid pressure via fluid injection can trigger seismic slip (Hubbert and Rubey, 1959; Healy et al., 1968;
Jaeger et al., 2007; Zoback and Gorelick, 2012; Walsh and Zoback, 2015; Walsh and Zoback, 2016).
Because of the critically stressed nature of the crust, a given fault will remain in a locked state as long
as the applied shear stress is lower than the strength of the contact (Hubbert and Rubey, 1959). The
critical shear stress is the product of the coefficient of friction and the effective normal stress given by
the difference between the applied normal and the pore pressure (Hubbert and Rubey, 1959; Raleigh et
al., 1976; Byerlee, 1978). Then, the critical shear stress on the earthquake fault under static friction is
given by the following expression:




$$\tau_c = \mu(\sigma_n - P_f)$$
$$P_f = P_0 + \Delta P$$     (1)

where $\tau_c$ is the critical shear stress (MPa), $\sigma_n$ is the normal stress (MPa), $P_f$ is the total pore pressure
(MPa), $P_0$ is the natural pore pressure (MPa), and $\Delta P$ is the increasing pore pressure (MPa) via fluid
injection.

5        In ambient conditions, the effective normal stress, which is oriented normal to the plane of the

fault, effectively clamps the fault closed and reduces the likelihood of slip occurring on the fault.
During fluid injection, as the pore fluid pressure increases, the effective normal stress decreases
proportionally; this reduction in the normal stress unclamps the fault and may result in slip along
preexisting subcritical ruptures (Jaeger et al., 2007; Langenbruch and Shapiro, 2015; Walsh and Zoback,
2015). Fluid injection in deep wells can trigger earthquakes when the injection causes the pore pressure
to increase near preexisting potentially active faults (Rutledge et al., 2004; Zoback and Gorelick, 2012;
Catalli et al., 2013). In these near-critical pressure conditions, relatively small perturbations to the
ambient pore fluid pressure conditions can and do trigger earthquakes; the Basel and Pohang events are
examples of earthquakes that were triggered by insignificant stress perturbations caused by fluid
injection (Terakawa et al., 2012; Walter et al., 2015; Kim et al., 2018; Woo et al., 2019).
**4 Present tectonic stress field in the Tangshan seismic region**
**4.1 Stress field inversion from earthquake focal mechanisms**
Michael (1987) developed a linearized stress field inversion scheme that can be solved by applying a
least-squares technique. Based on Michael's method, Vavryčuk (2014) proposed a new iterative stress
inversion technique (MATLAB software package STRESSINVERSE) that not only allows the user to
determine the stress and fault orientations, but also accurately calculates the shape ratio. Furthermore,
using the focal mechanism data, STRESSINVERSE allows the user to estimate the frictional
coefficients of the seismogenic faults. In this study, we used the STRESSINVERSE software package
to perform a crustal tectonic stress field inversion in the Tangshan seismic region. The shape ratio $R$
(Gephart and Forsyth, 1984) is expressed as:
$$R = \frac{\sigma_1 - \sigma_2}{\sigma_1 - \sigma_3}$$     (2)
where $\sigma_1$, $\sigma_2$, and $\sigma_3$ represent the maximum, intermediate and minimum principal stress, respectively.

28       To resolve the spatial variations in the inversion stress regime, we use the regime stress ratio ($RSR$)

parameter defined by Simpson (1997):
$$RSR = (n + 0.5) + (-1)^n (R - 0.5)$$     (3)
where $n$ is equal to 0 for normal faulting (NF), 1 for strike-slip faulting (SS), and 2 for reverse faulting
(TF), respectively (Hergert and Heidbach, 2011).

33       An advantage of the $RSR$ parameter is that it provides a continuous scale from pure normal

faulting ($RSR = 0.0$), normal/strike-slip faulting (0.5–1.5), pure strike-slip faulting ($RSR = 1.5$),
reverse/strike-slip faulting (1.5–2.5), and pure reverse faulting ($RSR = 2.5$) (Simpson, 1997).

36       Using full waveform data, Lin et al. (2017) determined the focal mechanisms of 918 earthquakes


($M_L \geq 2.5$) that occurred between January of 2010 and June of 2014 in North China using the FOCMEC
(Sonke, 2009) and TDMT_ISO (Dreger and Helmberger, 1993; Minson and Dreger, 2008) methods. In
this massive data set, they identified 572 especially robust focal mechanisms. In our study, we used 75
focal mechanisms data ($M_L \geq 2.5$) from the Lin et al. (2017) data set that sample the Tangshan seismic
region (with latitudes of N 38.8°–N 40.4° and longitudes of E 117.2°–E 119.8°) as the input data for
our crustal tectonic stress field inversion. Moreover, we also used 23 focal mechanisms ($M_L \geq 3.0$) from
earthquakes that occurred between November of 2006 and November of 2009 (Huang and Wan, 2015;
Fan et al., 2019) and between November of 2015 and March of 2019 (Yang et al., 2016; Feng et al.,
2019). Our total Tangshan seismic region data set is comprised of 98 focal mechanisms ($M_L \geq 2.5$) that
occurred during 2006–2019 (Fig. 2). The focal mechanisms, which summarize the prevailing sense of
slip during a seismic event, are generally classified as thrust faulting (TF), normal faulting (NF),
normal faulting with a strike-slip faulting component (NS), thrust faulting with a strike-slip faulting
component (TS), or pure strike-slip faulting (SS) (Zoback, 1992).
Previous studies show that there are spatial stress variations in the Tangshan seismic region (Feng
et al., 2019). In order to investigate the crustal tectonic stress field in this area, we divide the Tangshan
seismic region into 0.1°×0.1° bins, where each bin contains at least one earthquake. With a confidence
interval of 95%, the results of the inversion in each bin include the predominant maximum principal
compressive stress orientation ($\sigma_1$), the regime stress ratio (RSR), and the frictional coefficient ($\mu$).

## 4.2 Present tectonic stress field in the Tangshan seismic region

As shown in Fig. 3a, the Tangshan seismic region is characterized by local stress heterogeneity. The
maximum principal stress ($\sigma_1$) orientations of ENE-EW dominate the Tangshan seismic region, while
some WNW (~100°–112°) $\sigma_1$ orientations occur near the Lulong fault ($F_6$) in Luanxian County. Zhang
et al. (2008) suggests that the $\sigma_1$ axis has orientations of ~70°–80° (ENE), 91° (EW), and 91° (EW) in
the Tangshan, Ninghe, and northern Luanxian counties, respectively. While investigating the tectonic
field homogeneity in the Tangshan area, Yang et al. (2016) found that the $\sigma_1$ axes had orientations of
~87°–92° (ENE-EW) and 103° (WNW) near the Tangshan fault and the Lulong fault, respectively. Our
$\sigma_1$ axis results generally coincide with the those of previous studies (Zhang et al., 2008; Yang et al.,
2016). Furthermore, our regional tectonic stress field inversion, which was constrained using all 98
focal mechanisms (Fig. 3b), revealed that N83°E is the dominant $\sigma_1$ orientation in the Tangshan seismic
region; this result is also consistent with previous studies in our study area (Li et al., 1980; Huang and
Wan, 2015; Lin et al., 2017; Fan et al., 2019) and in northern China (Xu et al., 2008).
Additionally, Fig. 3a suggests that the predominant RSR values vary between 0.66 and 1.58; these
values coincide with a normal/strike-slip faulting stress regime. This stress regime is characterized by
significant strike-slip faulting in the western (e.g., in Tangshan, Fengnan, Fengrun, and Ninghe counties
with RSR values of ~1.10–1.60) and eastern (e.g., in Fu'ning, Changli, and Laoting counties with RSR
values of ~1.20–1.45) parts of the Tangshan seismic region. The prevailing stress regime in the central
Tangshan seismic region is characterized by normal faulting with a small component of strike-slip
faulting (e.g., in Tanghai, Luannan, and Luanxian counties with RSR values of ~0.55–0.85). These
stress regimes are consistent with both the fault rupturing that occurred during the 1976 Tangshan
earthquake sequence (Butler et al., 1979; Huang and Yeong, 1997; Wan et al., 2017) and the present
active features of the main seismogenic faults in this area (Jiang, 2006; Guo et al., 2011; Guo and Zhao,

3  2019).

Fig. 4 shows that the estimated friction coefficients near the main seismogenic faults mainly vary
between 0.4–0.6; the Tangshan fault belt, the Luanxian-Laoting fault, the Changli-Ninghe fault, the
Jiyunhe fault, and the Cangdong fault all have a friction coefficient of ~0.4. Byerlee (1978)
summarized numerous laboratory experiments on different rock types and stated that at elevated
effective normal stresses (<100 MPa), the corresponding friction coefficient fell in the range of
~0.6–1.0. Townend and Zoback (2000) suggested that the ratio of the maximum to minimum effective
stresses corresponds to friction coefficients ranging from 0.6 to 1.0; these values indicate a state of
crustal equilibrium. When it comes to assessing the fault slip potential, an empirical friction coefficient
of 0.6 is typically invoked as the critical value (Zoback and Healy, 1992; Zoback et al., 2003; Moeck et
al., 2009; Qin et al., 2015; Lund Snee and Zoback, 2016; Lee and Ong, 2018; Zhang and Ma, 2021).
However, because the prevailing friction coefficient in our study area (0.40) is lower than the empirical
critical value of 0.6, we infer the presence of some weaker seismogenic faults in the Tangshan seismic
region; this conclusion agrees with the low friction coefficients ($\mu = 0.21$–0.45) found in the Changli
area of the eastern Hebei province (Feng et al., 2017). Multiple studies have reported low friction
coefficients near strong earthquake seismogenic faults. For example, the San Andreas Fault system has
$\mu$ values of ~0.18–0.26 (Hickman and Zoback, 2004; Carpenter et al., 2012), the Yinxiu-Beichuan fault
(a branch of the Longmenshan fault zone in the eastern margin of the Tibetan Plateau) is characterized
by a $\mu$ value of 0.4 (Verberne and He, 2010), and the friction coefficients of ~0.2–0.5 are found in the
Yishu fault zone (a branch of the Tan-Lu fault zone in eastern China) (Li et al., 2019).
**4.3 Hydraulic fracturing measurements in the MTY EGS field**
Currently, there are no in situ stress measurements of the MYT EGS field. However, we performed
hydraulic fracturing at the Qian'An borehole (QABH, depth of 600 m), the Changli borehole (CLBH,
depth of 600 m), and the Luanxian borehole (LXBH, depth of 1000 m) from 2009 to 2013 in the
northern part of the MTY EGS field (Fig. 1). The QABH, CLBH, and LXBH boreholes are
approximately 70 km, 55 km, and 30 km away from the MTY EGS field, respectively (Fig.1). The
locations, rock types, and rock mass integrity of the QABH, CLBH, and LXBH boreholes are listed in
Table 1. The in situ stress measurements derived from our hydraulic fracturing tests at three boreholes
are shown in Table 2 (Tan et al., 2014; Tan et al., 2015; Niu et al., 2015; Feng et al., 2017; Feng et al.,
2019). Based on these in situ stress measurements, we can estimate the magnitude of the principal
stresses at shallow depths near the MTY EGS field.
Table 2 shows that the magnitudes of $\sigma_H$, $\sigma_h$ and $\sigma_v$ vary between 4.04–28.51 MPa, 3.75–19.46
MPa, and 1.79–24.44 MPa, respectively, over a depth range of 67.5–922.44 m. The magnitudes of the
horizontal principal stresses ($\sigma_H$ and $\sigma_h$) and the natural pore pressure ($P_0$) increase with depth (Fig. 5a).
From this data together, we used linear regressions to determine how principal stresses and the natural
pore pressure vary with depth:
$\sigma_H = 0.0278H + 5.33$      $R^2 = 0.6215$ (4)




$\sigma_h = 0.0183H + 2.69$     $R^2 = 0.7973$                                    (5)
$P_0 = 0.01H$           $R^2 = 0.9980$                                       (6)
$\sigma_v = 0.0265H$                                                  (7)
where $R^2$ is the correlation coefficient.
The linear gradients of $\sigma_H$ and $\sigma_h$ are 0.0278 MPa/m and 0.0183 MPa/m near the MTY EGS region,
respectively. These gradients are slightly larger than those found in northern China above a depth of
4000 m ($\sigma_H$ gradient of ~0.0229–0.0233 MPa/m and $\sigma_h$ gradient of ~0.0162–0.0170 MPa/m) (Yang et
al., 2012; Huang et al., 2013), suggesting that there is a higher stress accumulation in the Tangshan
seismic region than there is throughout the rest of northern China (Niu et al., 2015; Feng et al., 2019).
Because the three principal stresses have magnitudes of $\sigma_H > \sigma_v > \sigma_h$ below a depth of 328 m, we infer that
this area is dominated by a strike-slip faulting regime (Fig. 5a). The stress state at shallow depths (<
1000 m) is consistent with the results of the focal mechanism inversion performed at seismogenic
depths in the Tangshan seismic region (Fig. 3).
The measured $\sigma_H$ orientations vary from N47°W to N82°W (average of N65°±18°W) at borehole
QABH (depth range of 184.10–590.32 m), from N55°E to N82°W (average of N75°±20°E) at borehole
CLBH (depth range of 67.50–485.50 m), and from N60°W to N74°W (average of N68°±8°W) at
borehole LXBH (depth range of 340.00–922.44 m). The average $\sigma_H$ orientation for all three boreholes
at shallow depths (< 1000 m) is N80°W (or 100°), which is similar to the predominant $\sigma_1$ axis
orientation (N83°E) throughout the entire Tangshan seismic region (Fig. 3b, Fig. 5b).
Previous studies suggest that the customary vector approach, which involves averaging the
orientation of the maximum principal stresses, may yield unreasonable results and violate the tensorial
nature of the stress variable (Gao and Harrison, 2017). Using Eq. (A2) from Feng et al. (2020) and our
in situ stress data (Table 2), we first calculate the two-dimensional stress tensors at similar depths in the
$x$ (east) - $o$ - $y$ (north) coordinate system and then determine the tensorial mean of these stress states
(Table 3). Finally, we estimate the predominant orientation of the maximum horizontal principal stress
in different boreholes by applying the mean stress tensors to Eq. (6) from Feng et al. (2020). As shown
in Table 3, the predominant $\sigma_H$ orientations are N86°E at a depth interval of 206.50–212.92 m, N86°E
at a depth interval of 340.00–365.50 m, and N85°E at a depth interval of 468.68–485.50 m. These
estimated $\sigma_H$ orientations are consistent with the tectonic stress field generated by our focal mechanisms
inversion for the Tangshan area (N83°E) (Fig. 3b).
**5 Initial fault slip potential in Tangshan seismic region in the present stress field**
**5.1 Method for assessing fault slip potential**
We utilize the FSP v.1.0 software package to estimate the slip potential on the active faults throughout
the Tangshan seismic region. The FSP program allows for either a deterministic or probabilistic
geomechanical analysis of the fault slip potential. Both the deterministic and probabilistic
geomechanical models rely on several simplifying assumptions (Walsh et al., 2017): (1) the natural
pore pressure and stress tensor are uniform across the study area and linearly increase in magnitude
with depth, (2) one of the principal stress vectors is vertical, and (3) the stress state is determined by the


relative magnitude of the vertical stress vector (maximum, intermediate, or minimum, respectively).
The FSP tool allows the user to estimate the likelihood that the planar fault segments in question will
be critically stressed within a local stress field. When the ratio of the resolved shear stress to the normal
stress reaches a specific failure criterion (determined using the linearized Mohr-Coulomb failure
envelope), the fault becomes critically stressed (Lund Snee and Zoback, 2018). It should be noted that
the FSP program does not predict earthquakes. Instead, the FSP program assesses the cumulative
conditional probability of slip occurring on known faults, rather than quantifying the seismic hazard of
a given fault (Walsh et al., 2017).
Despite some limitations, FSP provides a forward looking probabilistic screening tool for known
faults near injection operations. Using the FSP tool, Walsh and Zoback (2016) calculated the
conditional probability of slip on mapped faults in response to injection-related increases in pore
pressure in north-central Oklahoma (USA), where widespread injection of produced saltwater has
triggered thousands of small to medium-sized earthquakes; Lund Snee and Zoback (2018) estimated
the potential for slip on mapped faults across the Permian Basin of west Texas in response to
injection-related pressure changes at depth that might be associated with future oil and gas
development activities in the region. Hennings et al. (2019) conducted a probabilistic assessment of
fault slip potential on the regionally mapped faults in the hydrocarbon-producing Fort Worth Basin of
north-central Texas, coinciding spatiotemporally with injection of 2 billion barrels of wastewater into
deep aquifers.
**5.2 Active faults in Tangshan seismic region**
Here, we use only publicly available information related to the most active faults in the Tangshan
seismic region. Based on the results of urban active fault explorations conducted in Tianjin City by the
Tianjin Earthquake Agency (Zheng et al., 2006; Chen et al., 2010; Liu et al., 2013; Yan et al., 2014) and
in the Hebei province of China conducted by the Hebei Earthquake Agency (Peng and Meng, 2017), we
collected location, length, strike, and dip information for the main active faults in the Tangshan seismic
region. The simplified strike data for various fault segments are shown in Fig. 6. This data set contains
53 fault segments, each defined by two connected coordinate points. The three-dimensional geometries
of the active faults used in the fault slip potential calculation are listed in Table 4.
**5.3 Deterministic geomechanical assessment of fault slip potential**
We first apply the deterministic geomechanical function of the FSP tool to estimate the slip potential of
the main active faults in the Tangshan seismic region in the absence of fluid injection. As shown in
Eqs.(7) and (8), the maximum ($\sigma_H$) and minimum ($\sigma_h$) horizontal stress gradients are 0.0278 MPa/m and
0.0183 MPa/m, respectively. The vertical stress ($\sigma_v$) gradient is 0.0265 MPa/m and the initial pore
pressure ($P_0$) gradient is taken as 0.01 MPa/m (Fig. 5). The reference depth for these calculations is
3965 m, which is the depth of the uppermost boundary of the MTY EGS. We used a critical friction
coefficient ($\mu$) value of 0.4 (Fig. 4). The orientation of the maximum principal stress , N83°E (Fig. 3b),
is also added to the stress database.
Fig. 7 shows the results of a deterministic geomechanical assessment of the fault pore pressure


required to generate fault slip across the Tangshan seismic region. We find that the active faults will not
all instantaneously slip in the present stress field and natural pore pressure conditions (Fig. 7a).
However, the deterministic pore pressures required to cause slip on each fault segment vary with the
different fault strikes (Fig. 7b). About 23% of the faults striking NE or WNW are likely to slip in
response to a small fluid pressure increase ($\Delta P$ =2.58–4.93 MPa); some of these more critical faults
include the $F_{4-1}$, $F_{4-3}$, and $F_{4-6}$ segments of the Tangshan fault belt ($\Delta P$ = 2.58–2.85 MPa), the $F_{5-3}$, $F_{5-4}$,
and $F_{5-8}$ segments of the Changli-Ninghe fault ($\Delta P$ = 3.25–4.93 MPa), and the $F_{17-1}$, $F_{17-2}$, and $F_{17-3}$
segments of the Lengkou fault ($\Delta P$ = 4.49–4.72 MPa) (Fig. 7c). Many (~49%) of the NE-ENE or
WNW striking faults are likely to slip in response to a modest pore pressure increase ($\Delta P$ = 5.40–10.70
MPa); some examples of these faults include the Yejituo fault ($F_3$) ($\Delta P$ of ~5.47–6.15 MPa), the Haihe
fault ($F_{12}$) ($\Delta P$ of ~6.67–10.70 MPa), and the Qinbei fault ($F_9$) ($\Delta P$ of ~5.41–6.39 MPa) (Fig. 7c).
Nearly 19% of the faults will likely slip at a large pore pressure perturbation ($\Delta P$ = 12.38–19.54 MPa);
an example of these less sensitive fault is the northwestern segment of the Baigezhuang fault ($F_8$) (Fig.
7c). The deterministic geomechanical assessment of the fault pore pressure required to generate slip are
listed in Table S1.

**5.4 Probabilistic analysis of fault slip potential**

Because the deterministic model ignores some uncertainties that are often present in the strike, dip,
ambient stress field, and the coefficient of friction (Walsh and Zoback, 2016; Lund Snee and Zoback,
2018), the deterministic geomechanical results are not entirely reliable. To minimize these uncertainties,
we use a probabilistic geomechanical function to estimate the fault slip potential (FSP) on each fault
segment using a Monte-Carlo-type analysis to randomly sample the specified uniform uncertainty
distributions for the input parameters (Lund Snee and Zoback, 2018). A Monte Carlo approach is useful
because it propagates the relevant uncertainties through the model, producing a distribution of pore
pressure values that may result in fault slip (Walsh et al., 2017).
Qin et al. (2014) suggested that the gradients of $\sigma_H$ and $\sigma_h$ with depth are 0.0328 MPa/m and
0.0221 MPa/m in and around the Beijing region, respectively. Huang et al. (2013) reported that the $\sigma_H$
and $\sigma_h$ gradients in the Zhangjiakou-Beijing-Bohai tectonic belt are approximately 0.0228 MPa/m and
0.0159 MPa/m, respectively. Considering that our study area is located in the eastern Beijing region
and in the southeastern section of the Zhangjiakou-Beijing-Bohai tectonic belt, we infer that the linear
$\sigma_H$ and $\sigma_h$ gradients near the MTY EGS field may vary between 0.0228–0.0328 MPa/m and
0.0159–0.0221 MPa/m, respectively.
Based on these assumptions, we can apply reasonable values and uncertainty ranges for the
gradients of $\sigma_H$ (0.0278±0.005 MPa/m) and $\sigma_h$ (0.0183±0.0024 MPa/m) with depth. The fault strike and
dip angles have uncertainties of ±5° and ±10°, respectively, the coefficient of friction on each fault
segment varies from 0.35 to 0.45 (average of 0.40±0.05), and the direction of the $\sigma_1$ axis varies from
66° (ENE) to 100° (WNW) (average of 83°±17°). An example of this type of analysis is shown in Fig.
S1 for the Tangshan fault belt. The distribution of pressures required to cause slip on fault $F_{4-1}$ is
evaluated by randomly sampling the uniform distributions (shown in red) of the input parameter
distributions for 1000 geomechanical models.


Fig. 8 shows the results of our probabilistic fault slip analysis in the absence of fluid injection for
2020 with respect to the locations of recent earthquakes (2009–2019) with the magnitudes of $M$ 1.0–4.9
in the Tangshan seismic region (National Earthquake Data Center, China). It is noteworthy that most of
these earthquakes have occurred on mapped faults with relatively higher FSP values; for example, the
Tangshan fault belt ($F_4$) has a 31%–41% probability of fault slip, the Jiyunhe fault ($F_2$) has a 27%–37%
probability of fault slip, and the northeastern segments of the Changli-Ninghe fault ($F_{5-7}$, $F_{5-8}$, $F_{5-9}$)
have a 23%–35% probability of fault slip (Fig. 8a). However, many earthquakes have also occurred on
mapped faults with lower FSP values; for example, the Lulong fault ($F_6$) and the northwestern end of
the Luanxian-Laoting fault ($F_{7-1}$) have only a 5% and 3% probability of fault slip, respectively (Fig.
8b).

11         As shown in Fig. S2, we find that the probability of the fault slip potential on mapped faults $F_{6-1}$,

$F_{6-2}$, $F_{7-1}$, and $F_{7-2}$ is very sensitive to the $\sigma_H$ azimuth. In the present stress field with a $\sigma_H$ orientation of
83°±17°, the FSP values on faults $F_{6-1}$, $F_{6-2}$, $F_{7-1}$, and $F_{7-2}$ are inconsistent with the high number of
earthquakes observed in the Lulong basin (Fig. 8b); from this observation, we conclude that local stress
field variations are responsible for these moderate-small events. If the $\sigma_H$ azimuth changes from
83°±17° to 55°±17° in the Lulong basin, the probability of fault slip on faults $F_{6-1}$ and $F_{6-2}$ increases to
32%–34% (Fig. 9a). Additionally, if the $\sigma_H$ azimuth changes from 83°±17° to 120°±17° in the Lulong
basin, the probability of fault slip on faults $F_{7-1}$ and $F_{7-2}$ increases to 24%–25% (Fig. 9b). Generally, the
results shown in Fig. 9 suggest that the complex local stress field in the Lulong basin heavily influences
the fault slip potential and the earthquake activity in this area. Using the focal mechanisms from the
1982 Lulong $M$ 6.2 earthquake and its aftershocks, Li et al. (2006) investigated the local stress field in
the Lulong Basin. They found that the maximum principal stress ($\sigma_1$) axis orientation changed to
N43°E in the northern part of the Lulong Basin; this orientation is distinctly different from the
dominant orientation of the regional tectonic stress field in the Tangshan seismic region (ENE-EW).
**6 Fault slip potential in the Tangshan seismic region due to fluid injection**
In this section, we perform a quantitative risk assessment of the MTY EGS field. In this assessment, we
use the FSP v.1.0 program to determine the probability of the fault slip potential (FSP) in response to a
fluid injected-related pore pressure increase via the Mohr-Coulomb mechanism.
**6.1 Hydrology model**
Injection of fluids into a porous medium causes an increase in pore pressure that decays exponentially
with radial distance from the injection source. This pressure change radiates away from the well
axisymmetrically as injection continues; as such, the model calculates a radially symmetric pressure
profile for each injection well at a given time using Eqs. (8) and (9) (Ferris et al., 1962; Bear, 1979;
Hsieh and Bredehoeft, 1981):
$$T\left(\frac{\partial^2 h}{\partial x^2}+\frac{\partial^2 h}{\partial y^2}\right) = S\frac{\partial h}{\partial t} - Q(t)\delta(x)\delta(y) \tag{8}$$
     $$\Delta P = rh$$


$$h(x,y,t) = \frac{Q(t)}{4\pi T} W(u)\left[\frac{(x^2 + y^2)S}{4Tt}\right]$$

$$W(u) = \int_u^\infty \frac{e^{-u}}{u} du \qquad (9)$$

$$u = R^2 \frac{S}{4Tt}$$

where $h$ is the vertically averaged buildup of hydraulic head above the initial head (m), $T$ is the principal value of the transmissivity (m²/s), $S$ is the storage coefficient, $Q(t)$ is the variable injection rate (L/s), $r$ is the specific weight of the fluid (N/m³), $\Delta P$ is the vertically averaged pressure increase (MPa), $W(u)$ is the well function, and $R$ is the radius distance away from the injection well (m).

These groundwater flow equations describe the two-dimensional (2D) radial flow in a vertically confined aquifer containing a variable injection rate well. The idealized model of the reservoir makes several simplifying assumptions to compute pressure buildup and the subsequent falloff caused by fluid injection (Ferris et al., 1962; Papadopulos, 1965; Bear, 1979; Hsieh and Bredehoeft, 1981; Walsh et al. 2017): (1) the porous medium is fully saturated and has a uniform pressure distribution, (2) the hydraulic head is the same everywhere before the injection, (3) injection wells are treated as point sources in the 2D grid, (4) the permeability and porosity are constant and isotropic, and (5) interacting pressure plumes are superimposed linearly. Using this hydrologic model, Hsieh and Bredehoeft (1981) approximated the pressure buildup in response to injection of fluid wastes into the fractured Precambrian crystalline bedrock beneath the Rocky Mountain Arsenal (RMA) near Denver triggered earthquakes in the 1960's, and their results showed that the increase of fluid pressure triggered the swarm of earthquakes at the RMA.

**6.2 Hydrological parameters and injection well conditions**

The necessary hydrological parameters are the injection formation thickness and the porosity and permeability of the injection layer. The aquifer thickness of the MTY EGS unit is 35 m (depth range of 3965–4000 m). Moreover, the MTY EGS field lies in the gneiss unit of the Bai'miaozi Series in the Dan'tazi Group of the Archaeozoic Erathem (Wang et al., 2013; Zhang et al., 2014), with an average pre-enhancement porosity of 6.9% (Zhou, 2003; Cao, 2016). The fractured reservoir permeability is closely related to the apertures of the fractures and the average spacing between fractures (Murphy et al.,1999). The actual fracture aperture mainly ranges from 0.05 mm to 2 mm, while the fracture spacing usually ranges from several meters to dozens of meters (Murphy et al., 1999; Sanyal and Butler, 2005). For a parallel fracture set, the average reservoir permeability theoretically ranges from 1 mD to 100,000 mD (Zeng et al., 2013). Based on data from the oil and gas industry, however, the fracture permeability following enhancement generally falls in the range of 1–100 mD (Sanyal and Butler, 2005; Zeng et al., 2013; Yue et al., 2015). Due to the lack of existing permeability measurements in the MTY EGS field, we must rely on a reasonable estimate of the fracture permeability. In this study, we assume that the average fracture permeability of the MTY EGS field in the presence of hydraulic stimulation is equal to 100 mD.

Five hypothetical injection wells with identical injection rates (W01, W02, W03, W04, and W05




in Fig. 6) are placed in the MTY EGS region. The injection well data describes the injection rate profile
of each well over time, from January $1^{st}$ of 2020 to December $31^{st}$ 2050. Evans et al. (2012)
determined many injection parameters for large induced earthquakes that were caused by fluid injection
in geothermal and $CO_2$ reservoirs in Europe. Their results showed that the circulation injection rates
associated with the largest magnitude events ranged from 18–120 L/s, (average of 51 L/s). In this study,
we used a fluid injection rate of 51 L/s (with a fluid density of 1000 kg/m³) to calculate the pore
pressure diffusion near the MTY EGS field.
**6.3 Probabilistic fault slip potential due to fluid injection**
Fig. 10 shows the fluid pressure perturbations from five injection wells linearly superposed onto the
mapped domain of the Tangshan seismic region in 2050. The increasing fluid pressure due to injection
into five wells varied between 0 and 11.56 MPa (Fig. 10a). Furthermore, the highest fluid pressure
increases occur within ~15–20 km from each injection well. However, beyond this range, the fluid
pressure perturbations induced by fluid injection quickly decay to zero (Fig. 10b). Figs. S3 and S4
show the fluid pressure perturbations from the five injection wells linearly superposed onto the mapped
domain in 2030 and 2040, respectively.
Fig. 11 presents the probabilistic FSP values in the presence of hypothetical fluid injection from
2030 to 2050. The detailed results are also listed in Table S2. A comparison of Figure 11 with the Fig. 8
(in 2020) suggests that the probabilistic fault slip potential on most of the active faults, such as the
Ji'yunhe fault ($F_2$), the Ye'jituuo fault ($F_3$), the Tangshan fault belt ($F_4$), the Lulong fault ($F_6$), and the
Xi'nanzhuang fault ($F_{11}$) do not exhibit any obvious changes from 2020 to 2050 because they are more
than 45 km away from the five injection wells (Table S2). For the faults that are within ~30–45 km of
the injection wells, such as the $F_{5-4}$ and $F_{5-5}$ segments of the Changli-Ninghe fault, the probabilistic FSP
values vary from 37% in 2020 to 38% in 2050 and from 18% in 2020 to 19% in 2050, respectively (Fig.
12a). Similarly, the probabilistic FSP values for the $F_{7-4}$ and $F_{7-5}$ segments of the Luanxian-Laoting
fault vary from 26% in 2020 to 27% in 2050 and from 32% in 2020 to 33% in 2050, respectively (Fig.
12b). Additionally, the probabilistic FSP values on the $F_{8-3}$ segment of the Bai'gezhuang fault changes
from 23% in 2020 to 25% in 2050 (Fig. 12c), while the FSP value does not change at all on the $F_{11-2}$
and $F_{11-3}$ segments of the Xi'nanzhuang fault (Fig. 12d). Overall, the hypothetical fluid injections only
weakly impact the probabilistic FSP values for the mapped faults at distances greater than ~30–45 km
away from the hypothetical injection wells in the MTY EGS field.
**7 Fault slip potential near the MTY EGS field due to fluid injection**
As mentioned previously, the FSP values for most active faults in the Tangshan seismic region
increased very little in response to sustained fluid injections from 2020 to 2050 because they were
located at distances greater than ~30–45 km away from the injection wells in the MTY EGS field.
Previous observations on injection-induced seismicity shows that large-scale, field-wide injections may
perturb faults and induce earthquakes at distances of ~30–40 km away from the wells (Keranen et al.,
2014; Goebel et al., 2017). Goebel and Brodsky (2018) suggested that fluid injection into sedimentary
rocks can lead to more large and distant earthquakes for a given volume of injection; this behavior





corresponds to a power law-like behavior for areas with distances from wells that exceed 15 km. As
such, we investigate the probabilistic FSP values for faults located within distances of ~15–20 km away
from the hypothetical injection wells in the MTY EGS field.

4       Previous work focusing on the seismic interpretation and the drilling strata for oil exploration have

revealed valuable information pertaining to the structures of the main boundary faults near the MTY
EGS field (≤ 20 km) (Zhou, 2003; Dong, 2011; Zhao, 2014). As shown in Fig. 13, the MTY EGS field
is located in the central Matouying uplift (II), where it is bounded to the north by the Bai'gezhuang
lower uplift (IV), to the south by the Shi'jiutuo depression (I), and to the northeast by the Laoting
depression (III and V). The boundary faults of these tectonic units (Fig. 13a), such as boundary faults
$F_{b1}$-$F_{b6}$ between the Matouying uplift and the Shi'jiatuo depression and faults $F_{b8}$-$F_{b14}$ between the
Matouying uplift and the Bai'gezhuang lower uplift, are mainly characterized as normal faults with
large dips (Fig. 13b). Based on these field studies, we have determined the locations, lengths, strikes,
and dips of the main boundary faults near the MTY EGS field. The various strikes of the 20 different
fault segments are shown in Fig. 13a. The three-dimensional geometries of these boundary faults that
are used to calculate the probabilistic FSP values are listed in Table 4. We utilize the FSP v.1.0 program
to estimate the probabilistic FSP values for these boundary faults using the same stress, hydrology, and
injection well conditions described previously.
Fig. 14 presents the probabilistic FSP values for the mapped faults near the MTY EGS field in
2020, 2030, 2040, and 2050. Fig. 15 shows the FSP changes that have occurred on certain main
boundary faults (e.g., $F_{b4}$-$F_{b6}$, $F_{b7}$, $F_{b8}$-$F_{b10}$, $F_{b11}$-$F_{b14}$, and $F_{b16}$-$F_{b20}$ segments) throughout the period of
fluid injection. The detailed results are listed in Table S3. Our results suggest that with continuing fluid
injection over time, the probabilistic FSP values on the boundary faults near the MTY EGS field will
progressively increase, especially for those faults with well distances of less than 15 km. Additionally,
the magnitude of the FSP changes vary with the fault strike and the distance from the injection wells.
For example, the FSP values for $F_{b11}$, $F_{b12}$, and $F_{b13}$ (NE orientation) closest to injection wells 03
and 04 (with well distances ≤ 6 km) vary from 38.5% in 2020 to 59.5% in 2050, from 29.5% in 2020 to
59.7% in 2050, and from 11.1% in 2020 to 35.1% in 2050, respectively (Fig. 15d). These faults have
the largest FSP changes between 2020 and 2050, with increases of 21%, 30.2%, and 24%, respectively.
However, the FSP values for faults $F_{b4}$ (NE orientation) and $F_{b5}$ (WNW orientation), which have
similar well distances of 6 km of from injection wells 01 and 02, have smaller increases, with FSP
values of 8.1% in 2020 and 25% in 2050 and 14.6% in 2020 and 30.5% in 2050, respectively (Fig. 15a).
The increase in the FSP values for faults $F_{b4}$ and $F_{b5}$ are 16.9% and 15.9%, respectively. For fault $F_{b7}$,
which is 7.5 km away from injection well 02, the FSP value varies from 35.5% in 2020 to 47.8% in
2050 (Fig. 15b)
Generally, the growth in the FSP values decays as the well distance increases. For example, faults
$F_{b7}$ and $F_{b14}$, which are ~6–10 km away from the injection wells, have FSP value increases of 11.8%
and 14.9% (Figs.15b and d). Faults $F_{b6}$, $F_{b9}$, $F_{b10}$, $F_{b15}$, $F_{b17}$, $F_{b18}$, $F_{b19}$, and $F_{b20}$, which are ~10–15 km
away from the injection wells, have FSP value increases that fall between 4.1%–8.6% (Figs.15a, c, e,
and f). Lastly, faults $F_{b8}$ and $F_{b16}$, which are ~15–20 km away from the injection wells, have FSP value
increases of 2.9% and 2.0% (Figs.15c and e). However, faults $F_{b1}$, $F_{b2}$, and $F_{b3}$, which have NE





orientations and well distances of ~6–10 km, have very small FSP value increases of 0.1%–2.0% in
2020 and 0.1%–3.6% in 2050. The lower FSP values on these faults likely indicate that faults with a
strike of NEN/NS experience additional fault stability and have higher fault strengths in the present
ambient stress field.
**8 Discussion**
**8.1 Effect of injection rate on the fault slip potential in the MTY EGS field**
The injection rates and volumes at single wells may be related to nearby earthquake activity (Walters et
al., 2015). Earthquakes are more commonly associated with injection wells with high fluid injection
rates (Weingarten et al., 2015). Furthermore, earthquakes tend to occur just after rapid increases in the
injection rate (Kim, 2013). The likelihood of triggering earthquakes depends largely on the rate at
which the pore pressure increases, rather than the absolute magnitude of the pore pressure (Alghannam
and Juanes, 2020). Moreover, high injection rates in neighboring wells can also cause a cumulative
effect in the form of a large pressure halo that could trigger slip on potentially active faults (Keranen et
al., 2014; Walters et al., 2015; Walsh and Zoback, 2015).
We calculated the FSP values for the fault segments that are closest to injection wells 01, 03, and
04 in 2030, 2040, and 2050 ($F_{b11}$, $F_{b12}$, $F_{b13}$, and $F_{b14}$) (Fig. 16), with different fluid injection rates
ranging from 0 to 120 L/s in 20 L/s increments. As shown in Fig. 16, we found that the FSP values for
these fault segments increase linearly with the fluid injection rate in 2030 (Fig. 16a), 2040 (Fig. 16b),
and 2050 (Fig. 16c); the regression coefficients $R^2$ vary between 0.961 and 0.999. Because the $F_{b13}$
fault segment has the smallest well distance of these four fault segments, this segment experiences the
largest increases in the FSP gradient (%) versus fluid injection rate (L/s); the gradient changes are
0.3857 in 2030, 0.5000 in 2040, and 0.5679 in 2050.
**8.2 Effect of permeability on the fault slip potential in the MTY EGS field**
As discussed previously, we calculated the FSP values for the $F_{b11}$, $F_{b12}$, $F_{b13}$, and $F_{b14}$ fault segments in
response to hypothetical fluid injection near the MTY EGS field, assuming an average permeability of
100 mD. We must assume that there is some uncertainty in this permeability estimate. As such, we
recalculated the probabilistic FSP values for fault segments $F_{b11}$, $F_{b12}$, $F_{b13}$, and $F_{b14}$ in 2030, 2040, and
2050, respectively, with permeability values ranging from 10 to 250 mD in 10 mD increments (Fig.

29  17).

We found that an enhanced permeability could weaken the FSP values for these four fault
segments in 2030, 2040, and 2050. Furthermore, the FSP values of the mapped faults decrease
exponentially with higher permeability values during fluid injection. Nevertheless, Cappa et al. (2018)
suggest that permeability enhancement has an important effect on the pressure diffusion and aseismic
slip growth during fluid injection. Their results reveal that a more pronounced permeability
enhancement results in a larger aseismic slip zone. Moreover, aseismic slip may play a significant role
in triggering distant earthquake sequences located outside the target reservoir (Wei et al., 2015). As
such, the permeability and aseismic slip zone should be considered when conducting the





seiongoingsmic hazard assessment of a given region due to fluid injection.
**8.3 Effect of thermoelastic stress on the fault instability in the MTY EGS field**
The influence of temperature has not been considered in the analyses although temperature-induced
stresses may play a significant role during EGS stimulation (Ghassemi and Tao, 2016). As a thermally
uncoupled case of heating of a half space, the temperature-induced horizontal stresses can be estimated
using the solution provided by Cheng (2016):

$$\Delta \sigma_{horizontal} = \alpha_d \frac{1-2\nu}{1-\nu} \Delta T \qquad (10)$$
$$\alpha_d = K\beta_d$$

where $\Delta\sigma_{horizontal}$ is the change in horizontal stress (MPa) with a change in temperature $\Delta T$ (°C), $\alpha_d$ is a
drained thermoelastic effective stress coefficient (MPa/°C), $\beta_d$ is a volumetric expansion coefficient
(°C$^{-1}$), $K$ is the bulk modulus of the reservoir rock (MPa), $\nu$ is Poisson's ratio.
It is assumed that the temperature drop is uniform throughout the reservoir (~3965–4000 m), and
the average temperature will decline by 6 °C during 20 years (Segall and Fitzgerald, 1998). The
average Young's modulus $E$ of the gneiss in the MTY EGS field is equal to 20 GPa and the Poisson's
ratio $\nu$ is equal to 0.23 (Li and Dong, 2013). Therefore, the bulk modulus $K$ [$K=E/3(1-2\nu)$] is equal to
12.35 GPa. A reasonable value for granitic gneiss can be calculated using $\beta_d$ of 2.4e$^{-5}$ °C$^{-1}$ (McTigue,
1990), resulting in the $\alpha_d$ of 0.296 MPa/°C. Then, the changes in horizontal stress is calculated for a
temperature drop of 6°C with a value of 1.25 MPa.
Taking faults $F_{b7}$, $F_{b11}$, $F_{b12}$, $F_{b13}$, and $F_{b14}$ as examples, we determined the fault stress state in the
Mohr's circles in 2040, when the average temperature hypothetically decline by 6°C throughout the
MTY geothermal reservoir. As shown in Fig.18, the black dot marks the traction of the fault instability
without the influence of thermoelastic stress, while the red dot marks the traction of the fault instability
with the effect of temperature-induced stress changes. Generally, the effect principal stresses $\sigma_H$ and $\sigma_h$
both decrease by 1.25 MPa at a depth of 3965 m. The decreasing thermoelastic stress on the selected
faults in 2040 shifts the Mohr's circles to the left. The temperature drop-induced stresses play a slight
effect on the fault instability on the faults $F_{b7}$, $F_{b11}$, $F_{b12}$, $F_{b13}$, and $F_{b14}$ under a strike-slip faulting stress
regime.
**8.4 The predicted maximum magnitude of injection-induced seismicity in MTY EGS field**
It is possible to estimate the maximum magnitude of earthquakes induced by fluid injection through
statistical, analytical, or hybrid forecasting methods (Gaucher et al., 2015; Tharaka et al., 2020). In this
study, our predictions are based on the recently developed and commonly accepted models proposed by
McGarr (2014) and Galis et al. (2017).
McGarr (2014) predicted the maximum magnitude of injection-induced earthquakes by simulating
a fully-saturated reservoir with critically stressed and ideally oriented faults in the vicinity of an
injection well. The model generates a linear relationship between the maximum magnitude and the net
injected volume ($\Delta V$):





$M_{max} = G\Delta V$  (11)
where $G$ is the modulus of rigidity (MPa). This model is often used to estimate the upper bound of the
seismic moment ($M_{max}$) for planned injection activities with a single well or a set of wells.
Galis et al. (2017) also proposed a quantitative physics-based model to account for specific
aspects of earthquake physics. The theoretical scaling relation between the largest magnitude of the
earthquakes ($M_0^{max}$) and the net injected volume ($\Delta V$) can be expressed as:

$$M_0^{max} = \gamma \Delta V^{3/2}$$
$$\gamma = \frac{0.4255}{\sqrt{\Delta\tau_0}}(\frac{K\mu}{h})^{3/2} \quad (12)$$

where $\Delta\tau_0$ is the background stress drop (MPa), $K$ is the bulk modulus of the reservoir rock (MPa), $\mu$ is
the friction coefficient, and $h$ is the reservoir thickness (m).
For the parameters used in Eq. (11), the average Young's modulus $E$ of the gneiss in the MTY
EGS field is equal to 20 GPa and the Poisson's ratio $v$ is equal to 0.23 (Li and Dong, 2013). Therefore,
the modulus of rigidity $G$ [$G=E/2(1+v)$] is equal to 8.13 GPa. In addition, for the parameters used in Eq.
(12), the bulk modulus $K$ is equal to 12.35 GPa, the background stress drop $\Delta\tau_0$ in the Tangshan seismic
region is about 6 MPa (Xiao et al., 1992), the friction coefficient $\mu$ in this study is 0.4, and the reservoir
thickness $h$ of the MTY EGS is equal to 35 m (~3965–4000 m).
Using these two models, we estimated the maximum magnitude of injection-induced seismic
events within well distances of ~10–15 km in response to a monthly injection volume time series in the
MTY EGS field (Fig. 19). Using common logarithm function, we then fit these calculated results to
obtain the relationship between the maximum moment magnitude ($M_w$) of injection-induced
earthquakes in response to a net injected volume ($\Delta V$) in the MTY ESG field:
$M_w = 0.67 \times lg^{\Delta V} + 0.66$  (13)
$M_w = 1.02 \times lg^{\Delta V} - 0.48$  (14)
where Eqs. (13) and (14) are the best-fitting models generated using the methods of McGarr (2014) and
Galis et al. (2017), respectively.
In these calculation, we used different monthly injection volume ($V$) ranging from $2.59 \times 10^3$ m³ to
$3.11 \times 10^5$ m³ and, corresponding to the fluid injected rate of ~ 1 L/s–120 L/s. In addition, we also
considered the net injected volume ($\Delta V$) as 10%$V$, 20%$V$, 30%$V$, and 40%$V$, respectively, due to the
fluid loss injected into the ESG reservoir (Table S4).
As shown in Fig. 19a, when the accumulated net injected volume is larger than 5000 L, the
predicted $M_w$ slowly increases from $M_w$ 3.1 to $M_w$ 3.6 using the McGarr model and from $M_w$ 3.3 to $M_w$
4.1 using the Galis model. The $M_w$ estimations for both models are relatively similar. The maximum
predicted $M_w$ for an injection-induced earthquake in the MTY EGS field under an assumption of 10%
fluid loss is $M_w$ 4.1.
For a fluid loss of 20%, when the accumulated net injected volume surpasses 5000 L, the
predicted $M_w$ slowly increases from $M_w$ 3.1 to $M_w$ 3.8 with the McGarr model and from $M_w$ 3.3 to $M_w$
4.4 with the Galis model. In this scenario, the maximum $M_w$ of an injection-induced earthquake in the
MTY EGS field is $M_w$ 4.4 (Fig. 19b).



When the fluid loss is increased to 30% and the accumulated net injected volume is larger than 5000 L, the predicted $M_w$ slowly increases from $M_w$ 3.2 to $M_w$ 3.9 using the McGarr model and from $M_w$ 3.5 to $M_w$ 4.6 using the Galis model. The maximum $M_w$ of an injection-induced earthquake is $M_w$ 4.6 in these circumstances (Fig. 19c).

As shown in Fig. 19d, when the accumulated net injected volume is larger than 5000 L and the fluid loss is 40%, the predicted $M_w$ slowly increases from $M_w$ 3.3 to $M_w$ 4.0 with the McGarr model and from $M_w$ 3.6 to $M_w$ 4.7 with the Galis model. For 40% fluid loss, the maximum $M_w$ of an injection-induced earthquake in the MTY EGS field is $M_w$ 4.7.

Previous studies indicate that 53 earthquakes ($M_w$ 2.0–2.7) were caused by fluid injection (injection volume of ~300–360 $m^3$/day) in the Renqiu oilfield of North China from August to November of 1986 (Fig. 20), and that 13 earthquakes ($M_w$ 2.0–3.5) were caused by fluid injection (injection volume of ~300–390 $m^3$/day) from February to June of 1987 (Fig. 20) (Zhao and Yang, 1990).

To verify our predicted models (Eqs. 13 and 14) for the maximum $M_w$ of fluid injection-induced earthquakes, we estimated the possible $M_w$ of the injection-induced earthquake in the Renqiu oilfield, which is located ~310 km away from the MTY EGS field. As shown in Fig. 20a, the predicted maximum $M_w$ of the injection-induced earthquakes that occurred between August and November of 1986 using the McGarr model vary between 2.6 and 3.0 with an injected fluid loss of less than 40%; these values are largely consistent with the observed earthquake magnitudes ($M_w$ 2.0–2.7). With the Galis model, the predicted maximum $M_w$ of the injection-induced earthquakes that occurred between February and June of 1987 vary between 2.5 and 3.3 (Fig.20b); these values are also consistent with the recorded magnitudes of the actual earthquakes ($M_w$ 2.0–3.5). Based on this analysis, we conclude that both of the predicted models shown in Eqs. (15) and (16) can be adopted to estimate the maximum moment magnitude of fluid-injected seismic events in the MTY EGS field in anticipation of the upcoming EGS exploitation. By comparison, we find that the maximum magnitudes of the injection-related seismicity estimated with the Galis model are slightly greater than the values by the McGarr model.

**9 Concluding Remarks**

Our conclusions are as follows:

(1) At shallow depths, the linearly increasing gradients of the maximum ($\sigma_H$) and minimum ($\sigma_h$) horizontal principal stresses near the MTY EGS field in the Tangshan seismic region of North China are 0.0278±0.005 MPa/m and 0.0183±0.0024 MPa/m, respectively. The $\sigma_H$ orientations vary from 66° to 100° (average of 83°±17°) near the MYT EGS field.

(2) In the Tangshan seismic region, most earthquakes ($M$ 1.0–4.9 from 2009 to 2019) have occurred on the faults that, in the present tectonic stress field, have relatively high fault slip potentials. For example, the Tangshan fault belt has FSP values ranging from 31% to 41%, and the Jiyunhe fault belt has FSP values ranging from 27% to 37%. However, many earthquakes, such as those on the Lulong fault and the northwestern end of the Luanxian–Laoting fault, have also occurred on faults with lower FSP values (3%–5%). The existence of these seismic events likely indicates that there is a local


stress field with a $\sigma_H$ orientation of ~43°–55° in the Lulong basin.
(3) The linearly superimposed fluid pressure perturbations induced by continuous injection at five
hypothetical injection wells in the MTY EGS field do not exceed 12 MPa from 2020 to 2050 (average
fluid injection rate of 51 L/s). The fault segments that experience the highest fluid pressure
perturbations are those with well distances of ~15–20 km. However, beyond this range, the fluid
pressure perturbations induced by fluid injection rapidly decay to zero.
(4) When the well distance exceeds ~30–45 km in the MTY EGS field, the probabilistic fault slip
potential on most of the active faults does not change from 2020 to 2050 in the Tangshan seismic
region. At well distances of less than 15 km in the MTY EGS field, the probabilistic fault slip potential
on most of the boundary faults increase with continuing fluid injection over time, especially on faults
with well distances of ~6–10 km. For example, fault segments $F_{b11}$, $F_{b12}$, and $F_{b13}$ between the
Bai'gezhuang lower uplift and the Matouying uplift have FSP values of 59.5%, 59.7%, and 35.1% in
2050, respectively.
(5) When we experiment with fluid injection rates (0 to 120 L/s) in our hypothetical wells, we find
that on fault segments $F_{b11}$, $F_{b12}$, $F_{b13}$, and $F_{b14}$, which are the fault segments with the shortest well
distances in the MTY EGS field, the probabilistic fault slip potential increases linearly with the fluid
injection rate. However, the FSP values on these faults decrease exponentially with increased unit
permeability during fluid injection.
(6) When the monthly injection volumes fall in the range of ~$2.59\times10^3$–$3.11\times10^5$ m$^3$ and the
injected fluid losses vary between ~10%–40%, the predicted maximum moment magnitude of an
injection-induced earthquake is $M_w$ 4.1–4.7 in the MTY EGS field.
(7) We show how the FSP software package can be used as a quantitative screening tool to
estimate the fault slip potential in a region with some uncertainties of the ambient stress field, and to
assess the reactivation potential on these faults of presumably higher criticality in response to fluid
injection. The case study of the MTY EGS field has important implications for the deep geothermal
exploitation in China, especially for Gonghe EGS (in Qinghai province) and Xiong'an New Area (in
Hebei province) geothermal reservoirs that are close to the Quaternary active faults. Ongoing injection
operations in the regions should be conducted with these understandings in mind.
*Author contributions.* CJ.F conceptualised the research, performed the tectonic stress field and fault slip
potential analysis, and wrote the manuscript. SH.Z conceptualised the research, and provided the FSP
1.0 software package. GL.G implemented the integrated seismic and geological data interpretation of
the Matouying uplift and its vicinity. Both DS.S and CX.T analyzed in situ stress measurements in the
Tangshan seismic region. SY.Z estimated the maximum magnitude of injeciton-induced seismicity in
MTY EGS field. XD.M defined the research structure and reviewed the manuscript.
*Competing interests.* The contact author has declared that neither they nor their co-authors have any
competing interests.
*Acknowledgments.* The authors wish to thank professor Qunce Chen (Institute of Geomechanics, CAGS)



for providing in situ stress results measured at Luanxian borehole in Tangshan seismic region. We
would like to thank assistant professors Weifeng Sun, Peng Zhang and Bangshen Qi (Institute of
Geomechanics, CAGS) for conducting in situ stress measurements at Qian'an and Changli boreholes in
Tangshan seismic region. We also thank LetPub (www.letpub.com) for linguistic assistance and
pre-submission expert review.
*Financial support.* This research has been supported by the China Geological Survey (grant no.
DD20190317).

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

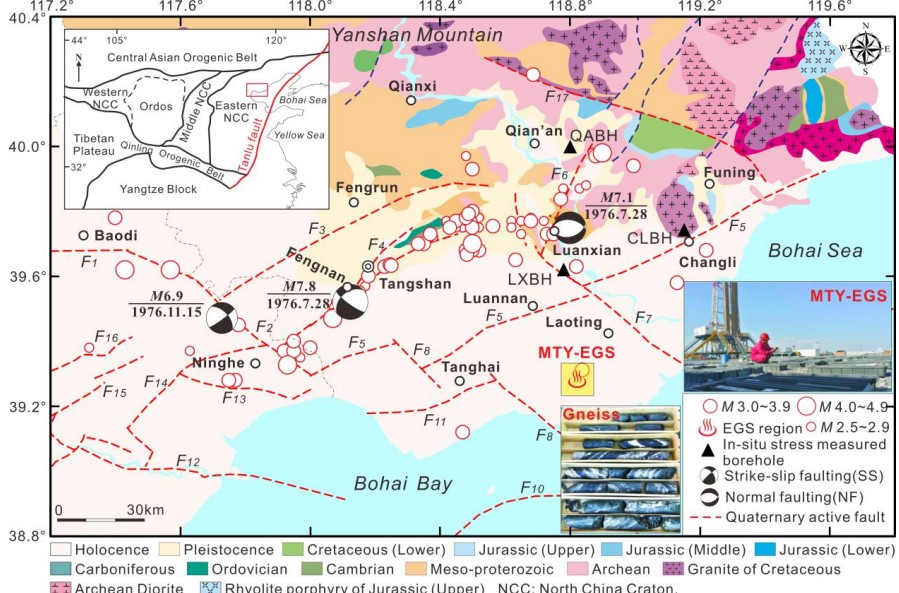

**Figure 1.** The geological-seismogenic structures and the earthquakes ($M \geq 2.5$) from 2006 to 2019 in the Tangshan
seismic region, North China (modified after Feng et al. (2019)). The earthquakes are gathered from the National
Earthquake Data Center (NEDC), China (https://data.earthquake.cn). Focal mechanisms of the Tangshan $M$ 7.8
(7.28/1976), Luanxian $M$ 7.1 (7.28/1976), and Ninghe $M$ 6.9 (11.15/1976) earthquakes were determined by Huang
and Yeong (1997). The tectonic faults in the Tangshan seismic region are as follows (Guo et al. 2011): $F_1$-Baodi
fault; $F_2$-Jiyunhe fault; $F_3$-Yejituo fault; $F_4$-Tangshan fault belt; $F_5$-Changli-Ninghe fault; $F_6$-Lulong fault;
$F_7$-Luanxian-Laoting fault; $F_8$-Baigezhuang fault; $F_9$-Qinbei fault; $F_{11}$-Xi'nanzhuang fault; $F_{12}$-Haihe fault;
$F_{13}$-Hangu fault; $F_{14}$-Cangdong fault; $F_{17}$-Lengkou fault. The yellow square denotes the Matouying (MTY)
enhanced geothermal system (EGS) field being approximately 90 km away from the Tangshan City in Hebei
province, China (Qi et al., 2020; Zhang et al., 2020).

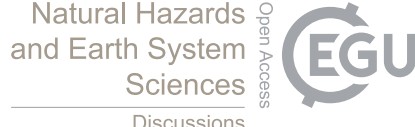

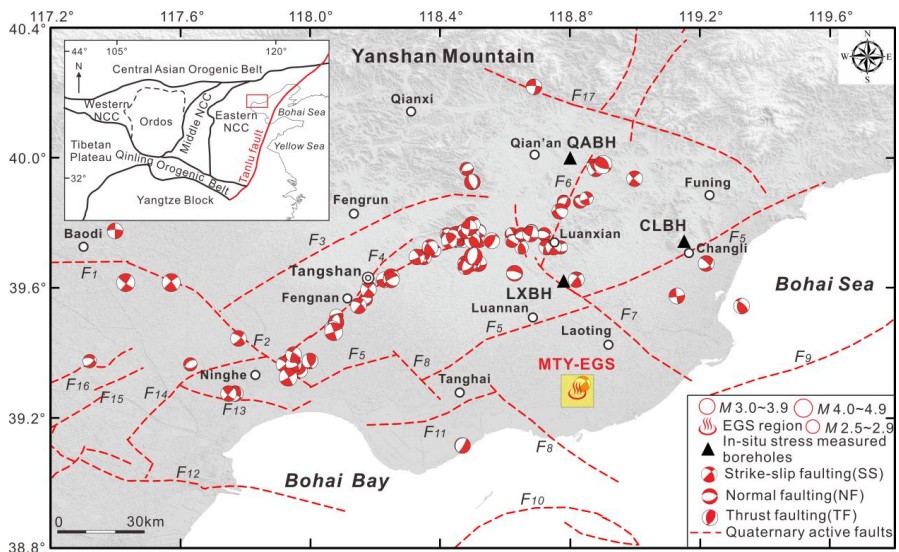

**Figure 2.** Focal mechanisms from 98 earthquakes ($M \geq 2.5$) from 2006 to 2019 in the Tangshan seismic region (gathered from Lin et al. (2017)) and in situ stress measured boreholes (black triangle) by hydraulic fracturing method on the north of the MTY EGS field (Tan et al., 2014; Niu et al., 2015; Feng et al., 2019).

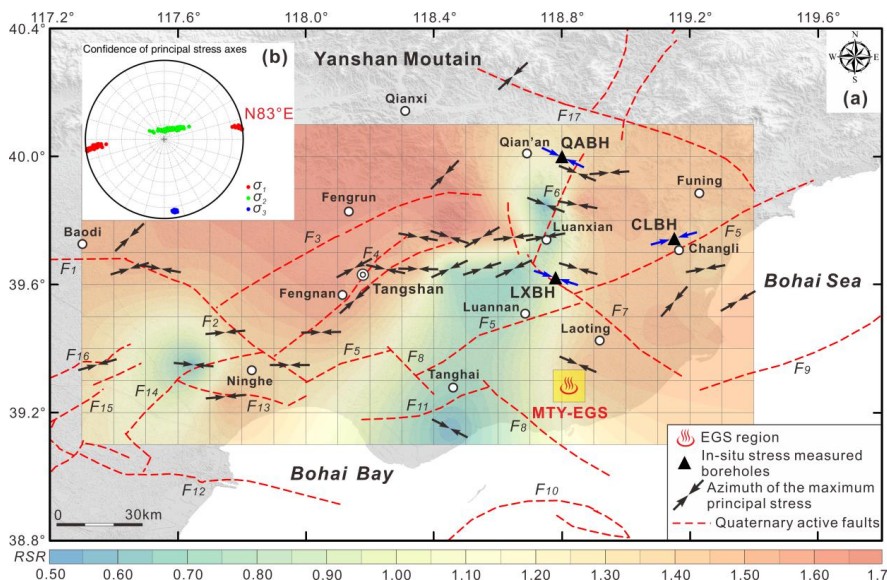

**Figure 3.** Crustal stress field in the Tangshan seismic region, as determined by inversion of 98 focal mechanisms shown in Fig.2 using STRESSINVERSE software package (Vavryčuk, 2014). **(a)** The results of the inversion in each bin (0.1°×0.1°) include the predominant maximum principal compressive stress orientation ($\sigma_1$) and the regime stress ratio (*RSR*). **(b)** The dominant $\sigma_1$ orientation in the Tangshan seismic region was constrained using all 98 focal mechanisms with a confidence interval of 95%.


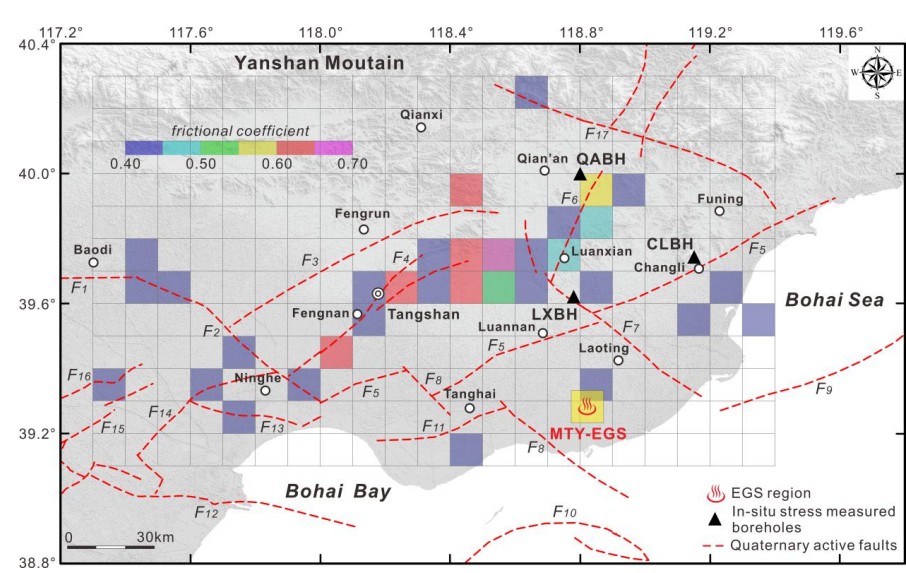

**Figure 4.** The critical coefficient of friction of the main seismic faults in the Tangshan seismic region, as determined by inversion of 98 focal mechanisms (seen in Figure 2) using STRESSINVERSE software package (Vavryčuk, 2014).

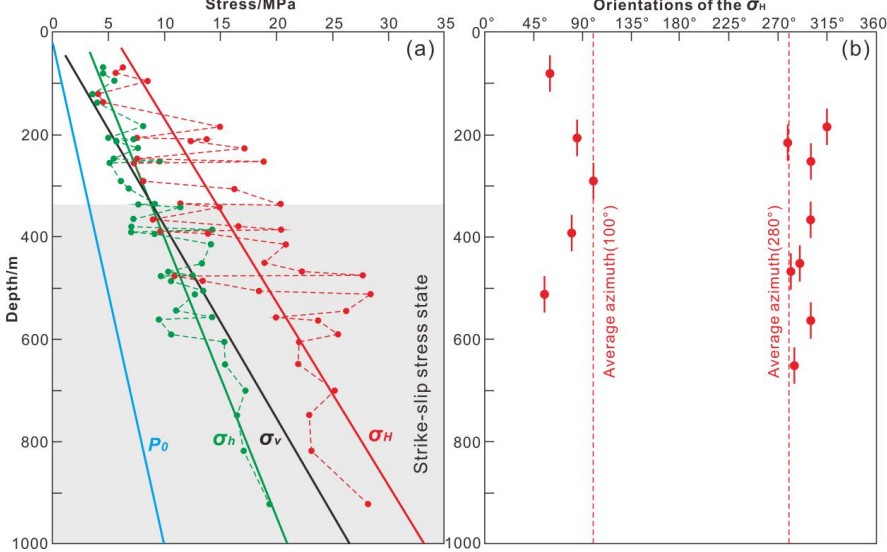

**Figure 5.** Hydraulic fracturing in situ stress results at shallow depths (< 1000 m) near the MTY EGS field in Tangshan seismic region. **(a)** The magnitudes of three principal stresses with the depth. **(b)** The orientations of the maximum horizontal principal stress ($\sigma_H$).

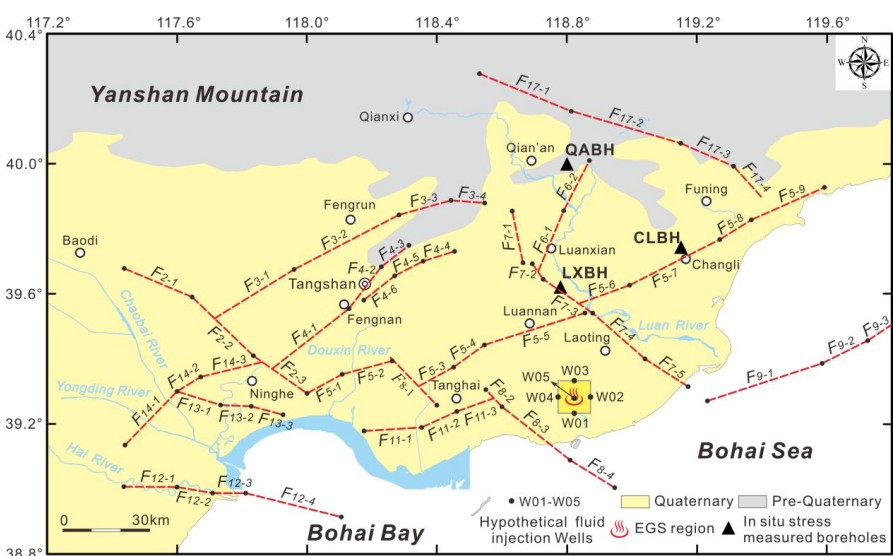

2 **Figure 6.** Simplified-segmental faults with different strikes in the Tangshan seismic region. There are total 53

3 segments used for calculating the fault slip potential (FSP).

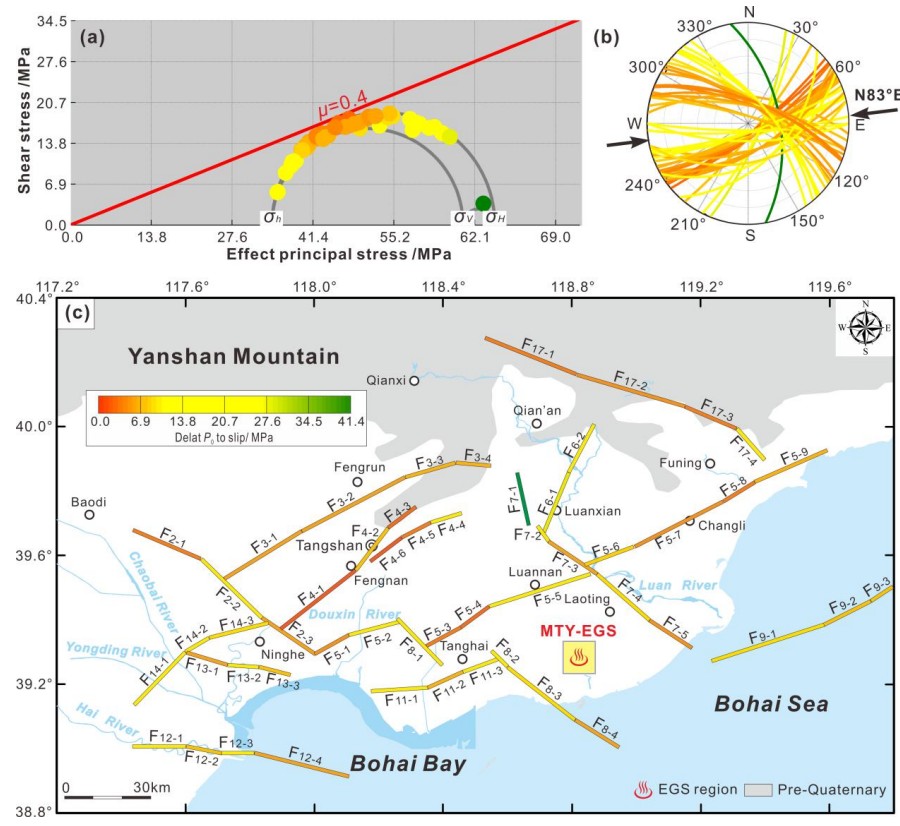

**Figure 7.** The results of a deterministic geomechanical assessment of fault pore pressure to slip across the
Tangshan seismic region. **(a)** The faults shown on a Mohr diagram, with effective compressive stress on the
horizontal axis and shear stress on the vertical axis; Principal stresses are labeled in black, and the frictional slip
line is shown in red. Faults are colored by their horizontal distance to slip in MPa (according to the color scale). **(b)**
Fault normal orientations plotted on a lower hemisphere equal angle stereonet as arcs; The azimuth of maximum
horizontal compression is shown with black arrows. **(c)** The same faults mapped and colored by deterministic fluid
pore pressure to slip.

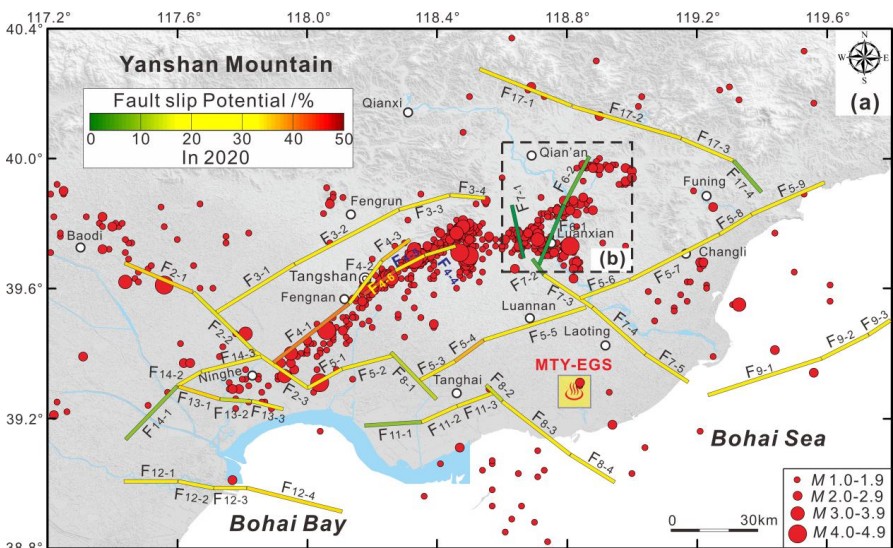

**Figure 8.** The probabilistic fault slip potential on the mapped faults without any fluid pressure perturbation in 2020
and the recent earthquakes (2009-2019) with the magnitude of *M* 1.0-4.9 in Tangshan seismic region (National
Earthquake Data Center, China).

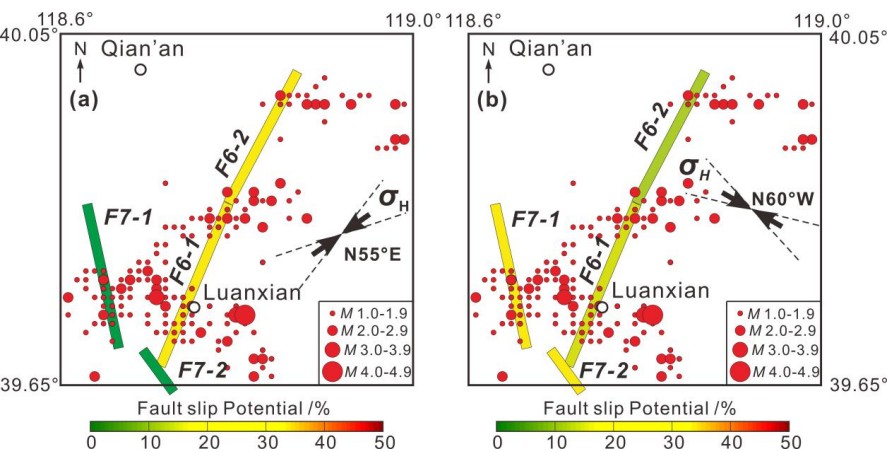

**Figure 9.** The probabilistic fault slip potential on the faults $F_{6-1}$, $F_{6-2}$, $F_{7-1}$, and $F_{7-2}$ in Lulong basin and the $\sigma_1$
azimuth of the hypothetic local stress field. **(a)** The $\sigma_1$ is oriented at 55°±17°. **(b)** The $\sigma_1$ is oriented at 120°±17°.



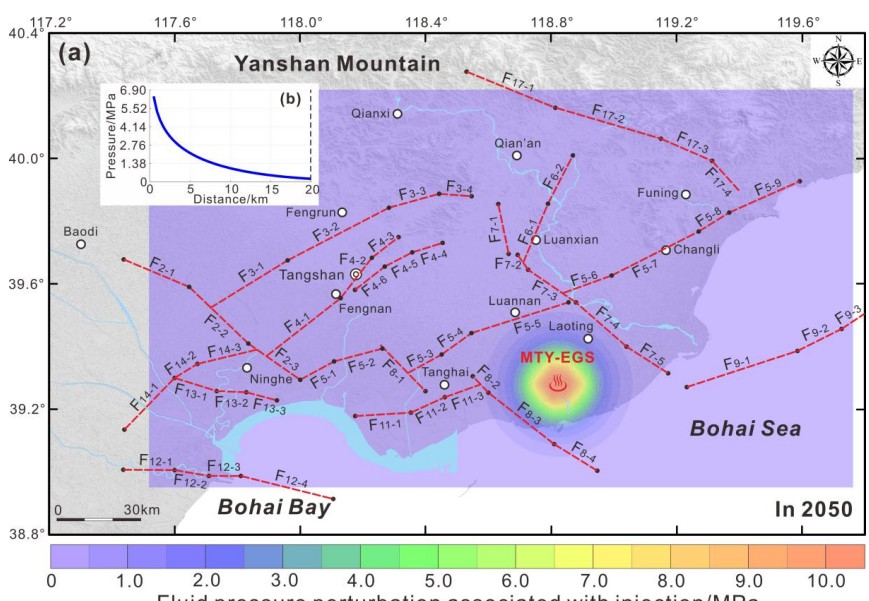

**Figure 10. (a)** Fluid pressure perturbations from five injection wells linearly superposed onto the mapped domain
of Tangshan seismic region in 2050. **(b)** Increase in fluid pore pressure above natural levels due to fluid injection in
each of the five wells as a function of distance in model year 2050.

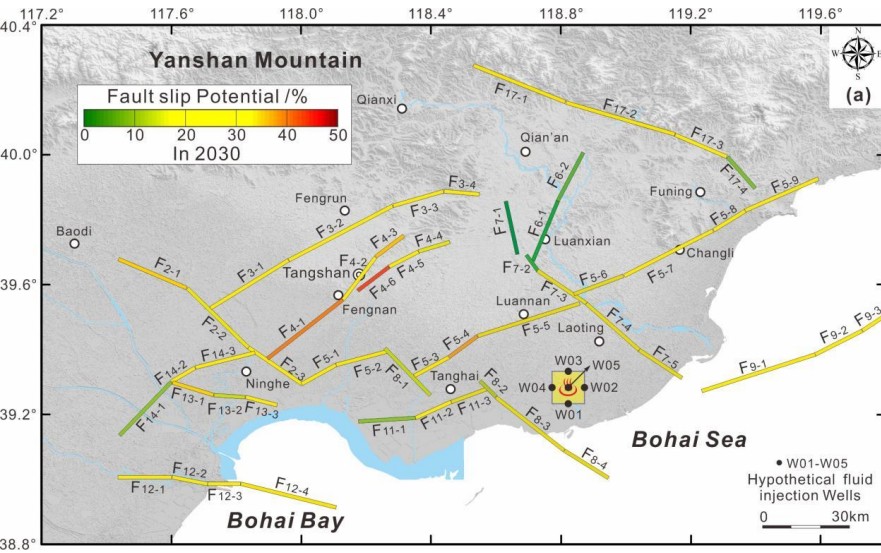

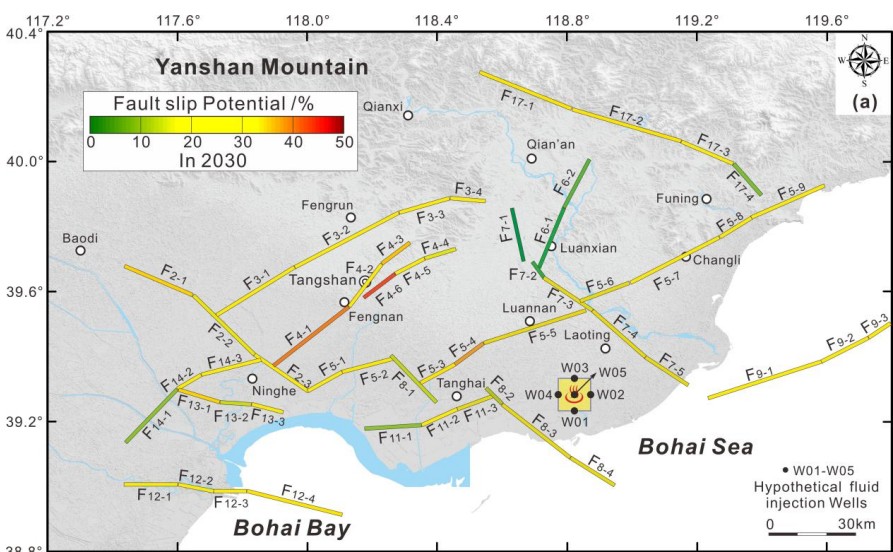

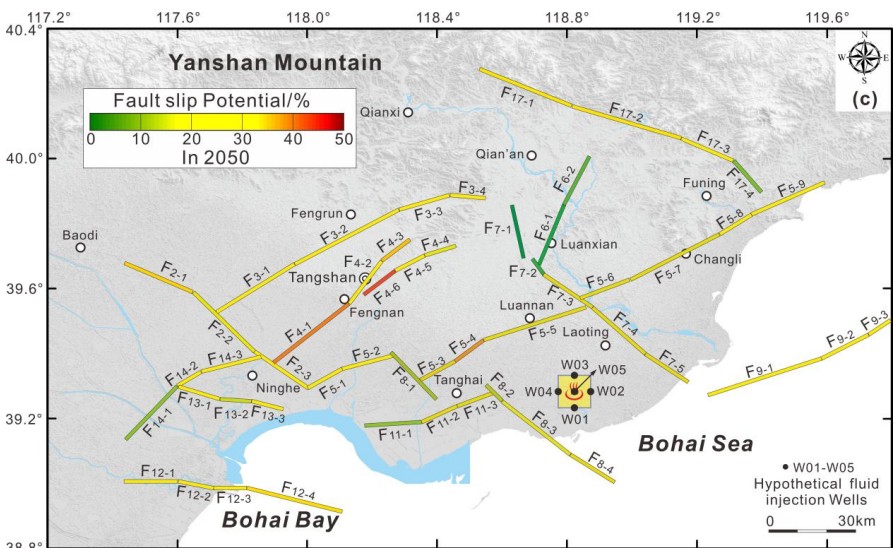

**Figure 11.** The probabilistic fault slip potential on the mapped faults in response to the hypothetic fluid injection in the MTY EGS field in 2030 **(a)**, 2040 **(b)** and 2050 **(c)**, respectively.



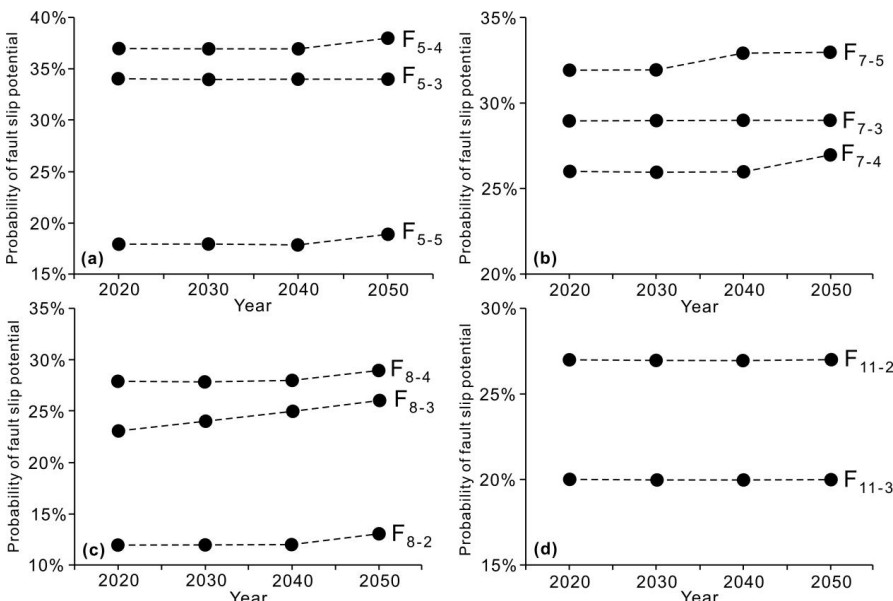

**Figure 12.** Changes of the probabilistic fault slip potential on the mapped faults within a range of ~30-45 km away
from the injected wells in the MTY EGS field from 2020 to 2050. **(a)** The FSP values on the southwestern segment
of the Changli-Ninghe fault. **(b)** The FSP values on the southeastern segment of the Luanxian-Laoting fault zone.
**(c)** The FSP values on the Baigezhuang fault. **(d)** The FSP values on the Xi'nanzhuang fault.

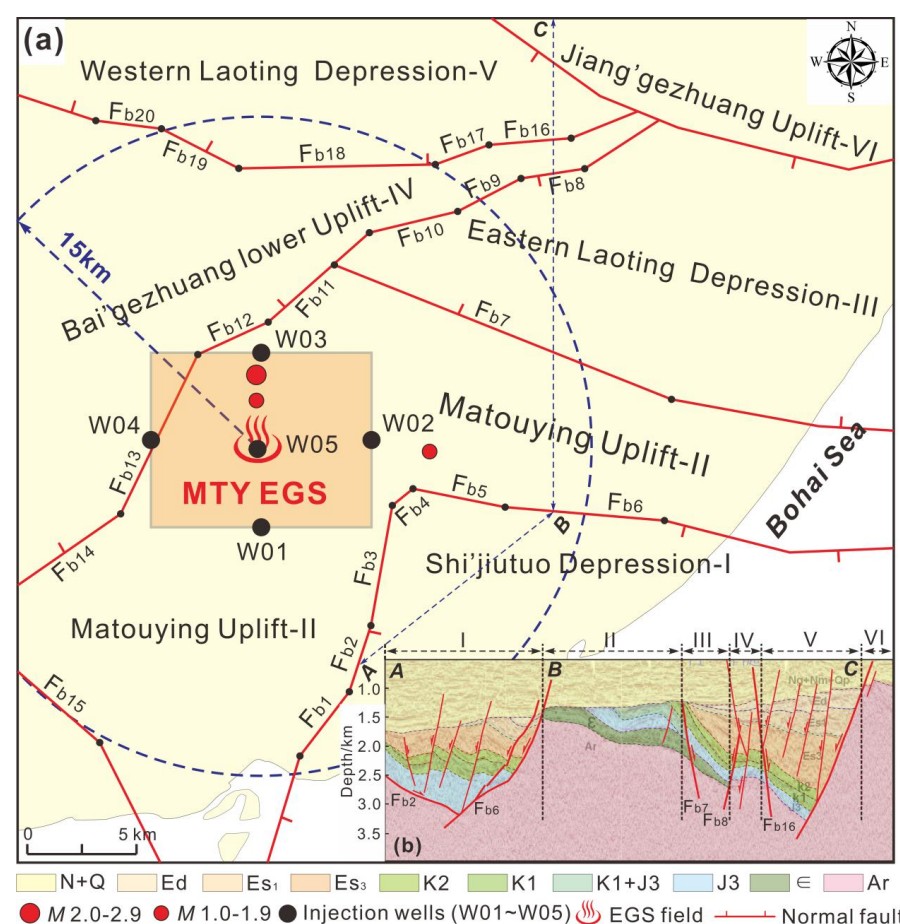

**Figure 13. (a)** Geological structures and active faults within a range of ~15-20 km away from the MTY EGS field.

**(b)** The structural framework of the Matouying uplift and its vicinity from integrated seismic and geological data

interpretation.

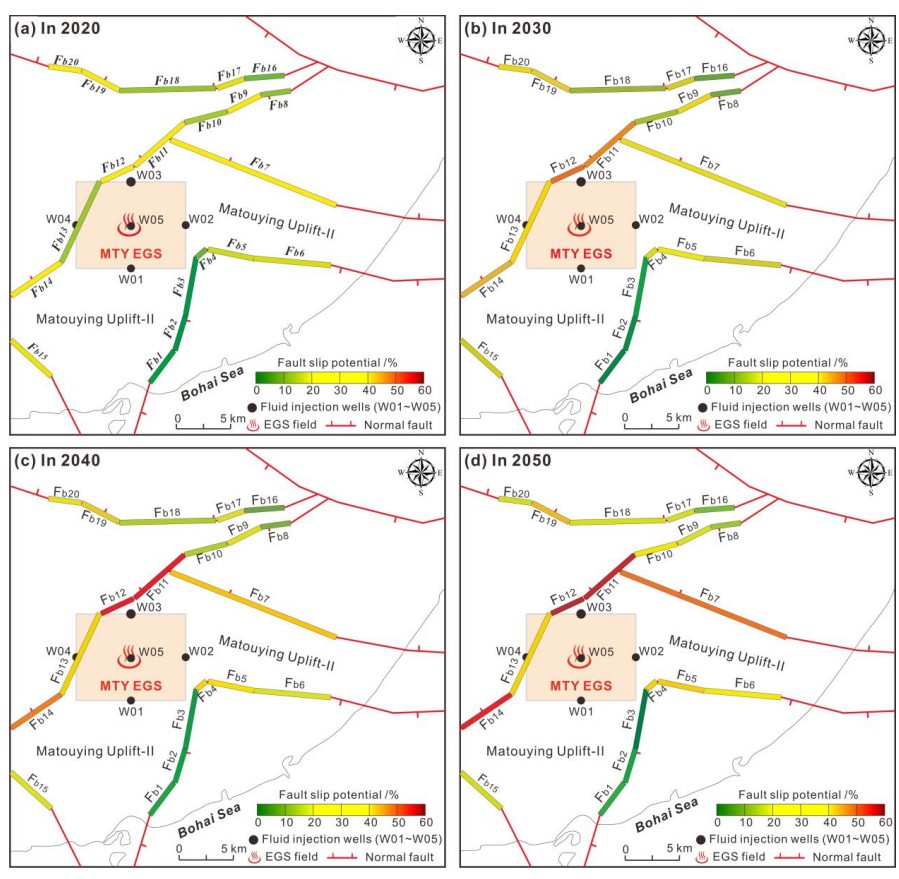

**Figure 14.** The probabilistic fault slip potential on the mapped faults within a range of ~15-20 km away from the MTY EGS field in response ot the hypothetic fluid injection in 2020 **(a)**, 2030 **(b)**, 2040 **(c)** and 2050 **(d)**, respectively.



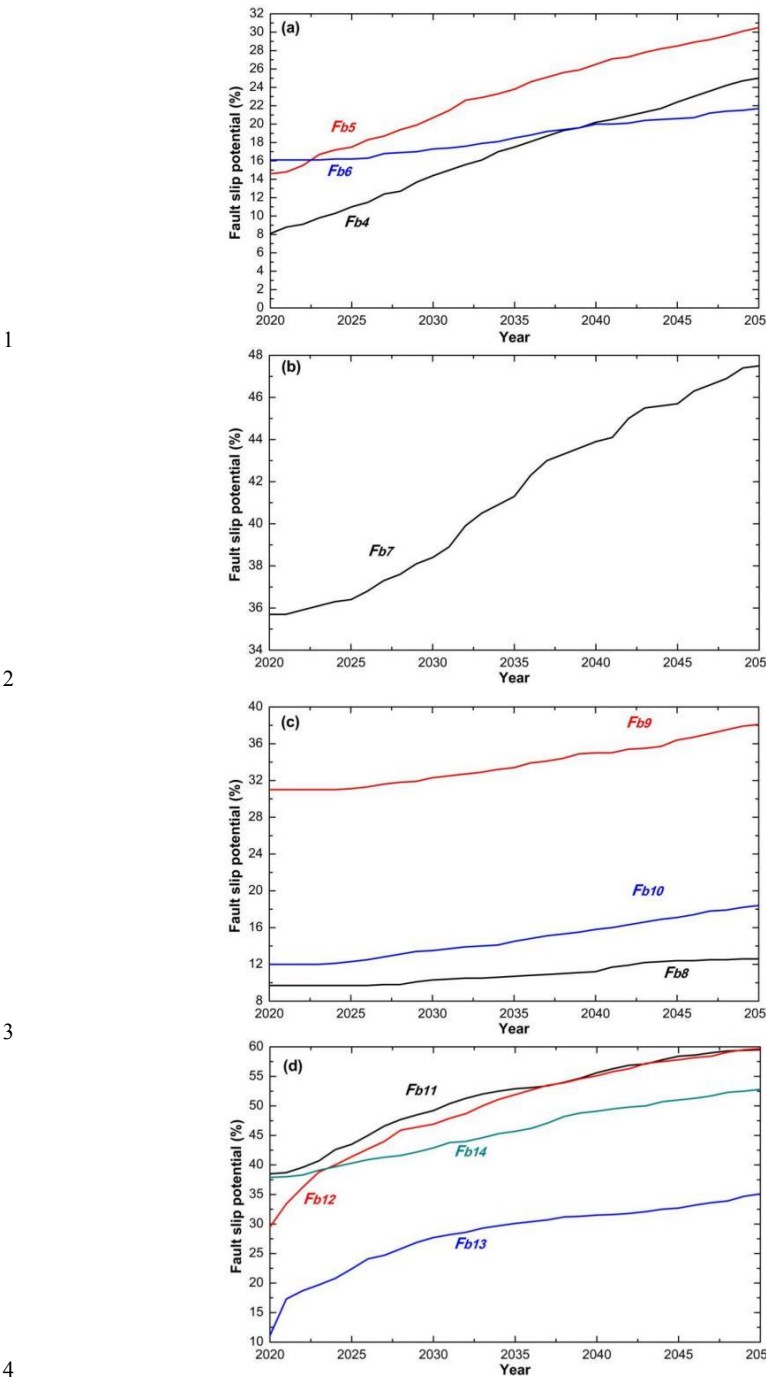

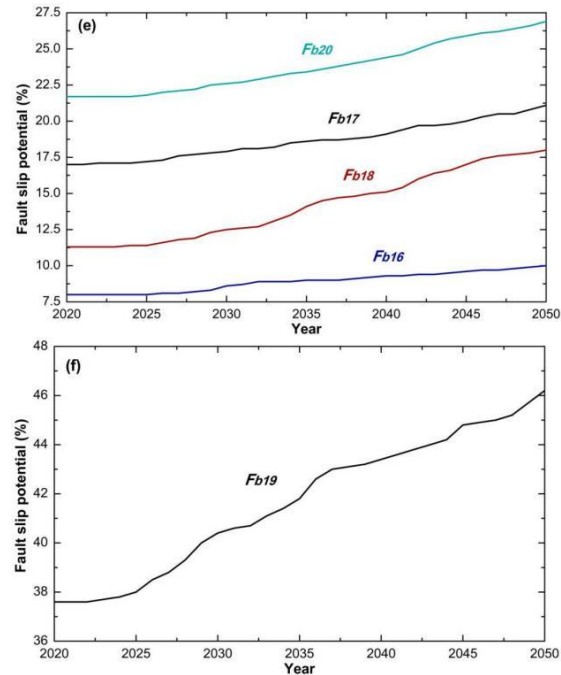

3  **Figure 15.** Changes of the probabilistic fault slip potential on the mapped faults within a range of ~15-20 km away

4  from the injected wells in response to the hypothetic fluid injection in the MTY EGS field from 2020 to 2050.



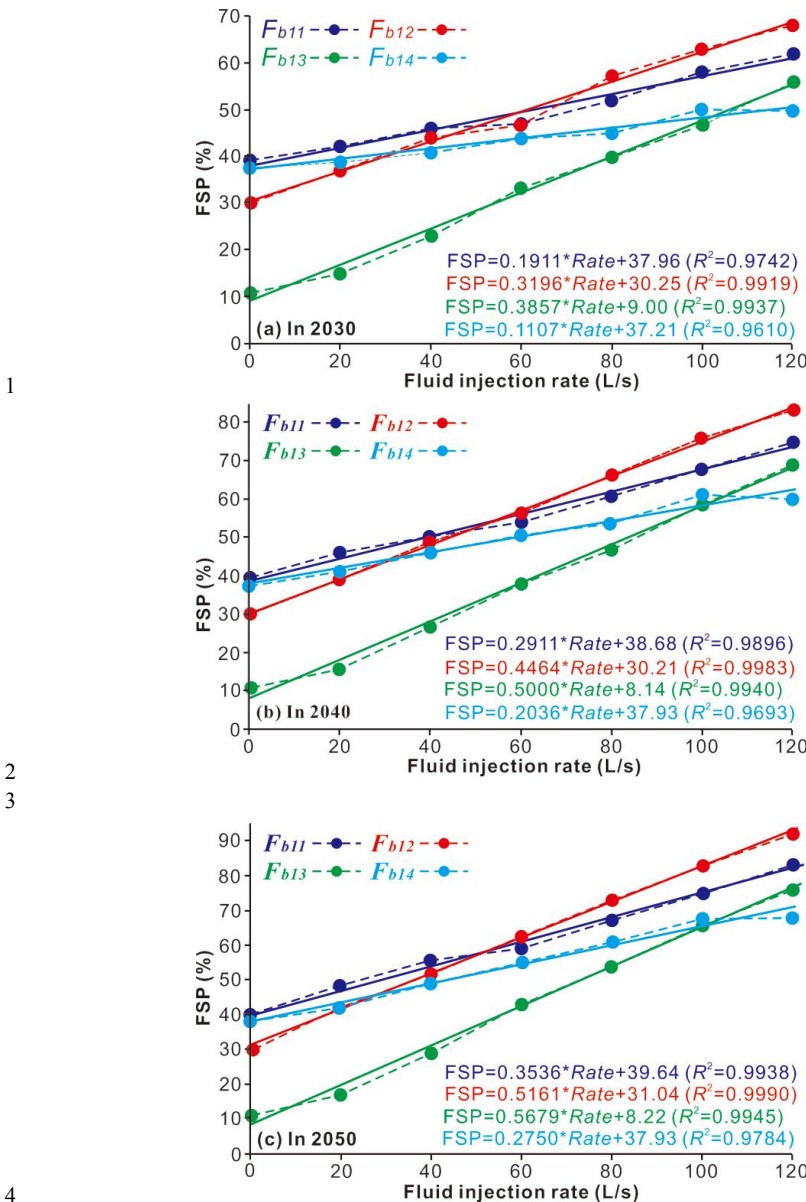

**Figure 16.** The effect of injection rate on probabilistic fault slip potential on the mapped faults $F_{b11}$, $F_{b12}$, $F_{b12}$, and $F_{b14}$, within a range of ~ 6-10 km away from the MTY EGS field.




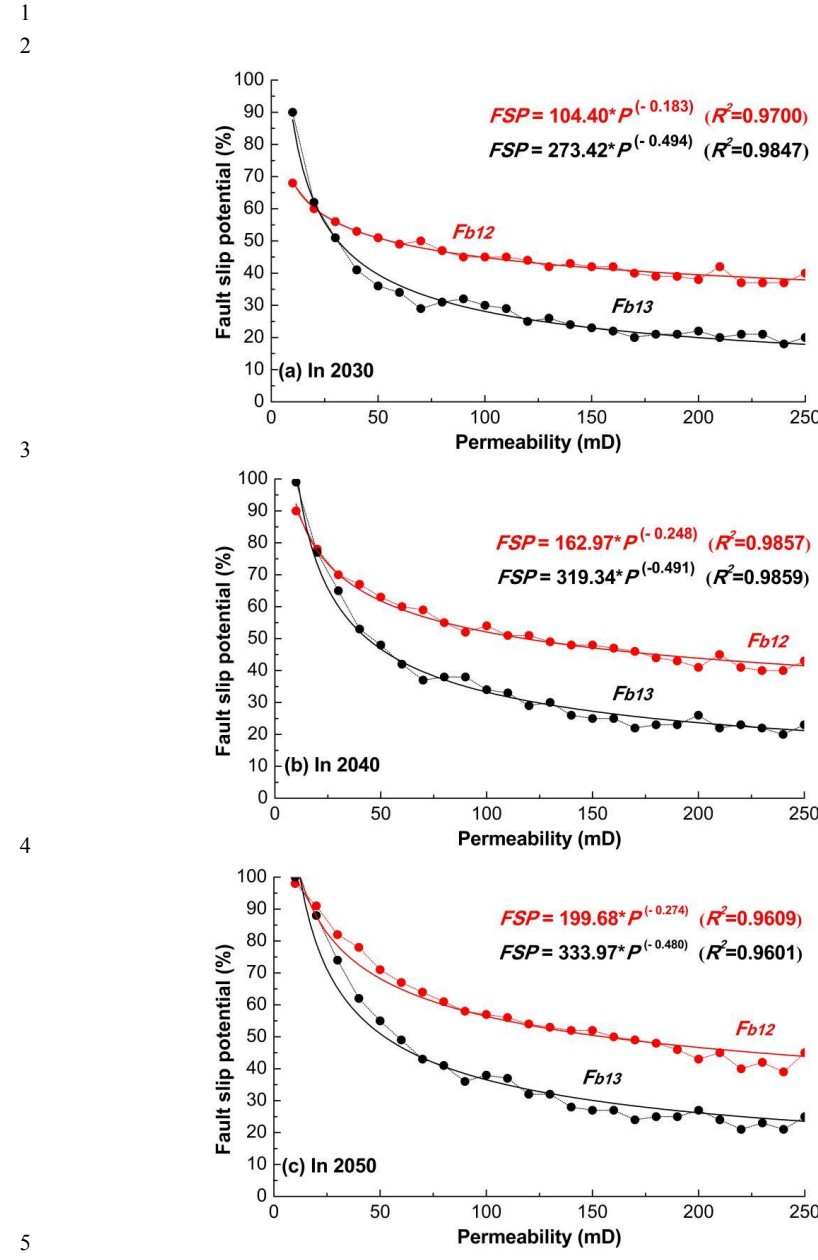

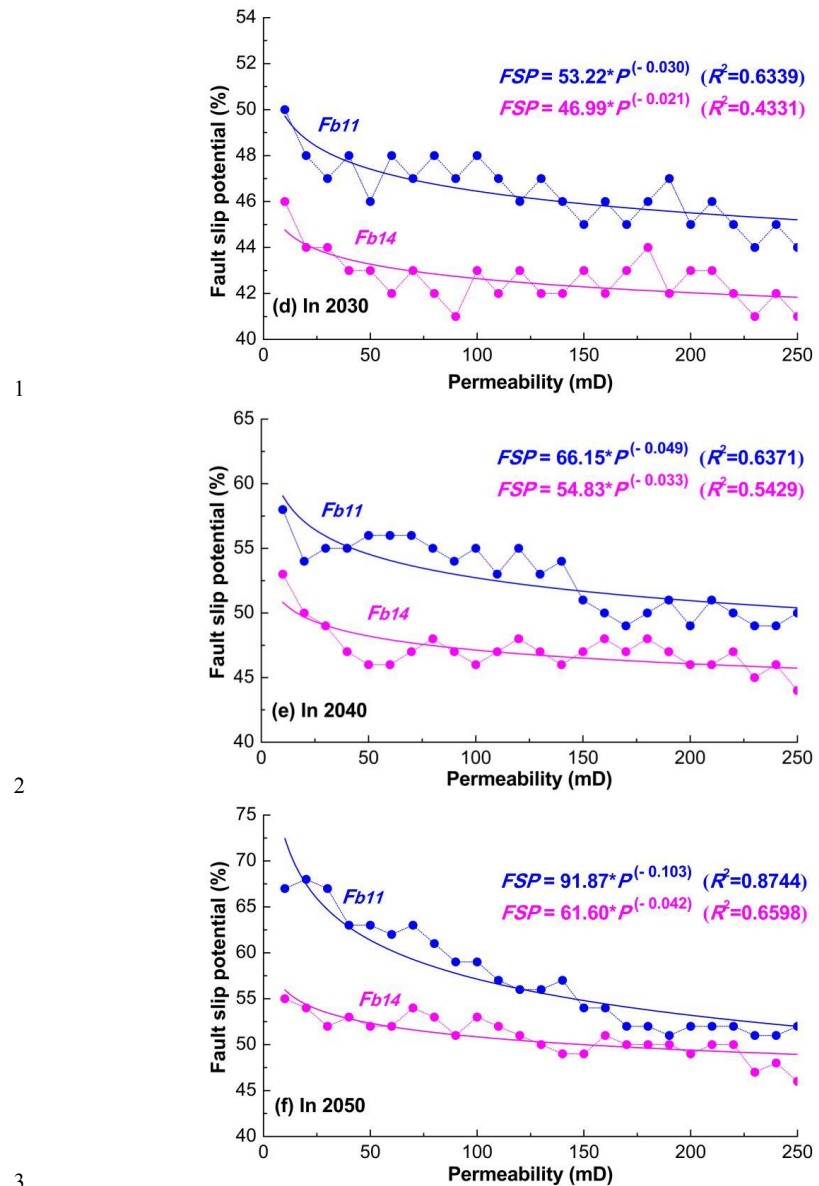

**Figuer 17.** The effect of permeability on probabilistic fault slip potential on the mapped faults $F_{b11}$, $F_{b12}$, $F_{b12}$, and $F_{b14}$, within a range of ~ 6-10 km away from the MTY EGS field.







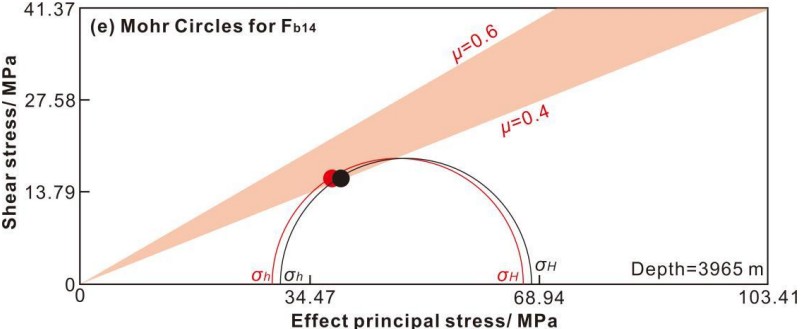

**Figure 18:** The effect of thermoelastic stress on fault instability of the mapped faults $F_{b11}$, $F_{b12}$, $F_{b12}$, $F_{b13}$ and $F_{b14}$,
within a range of ~ 6-10 km away from the MTY EGS field. The changes in horizontal stress is calculated for a
temperature drop of 6°C with a value of 1.25 MPa. The black dot marks the traction of fault instability (a
horizontal distance to slip in MPa) without the influence of thermoelastic stress, while the red dot marks the
traction of the fault instability with the effect of temperature-induced stress changes.

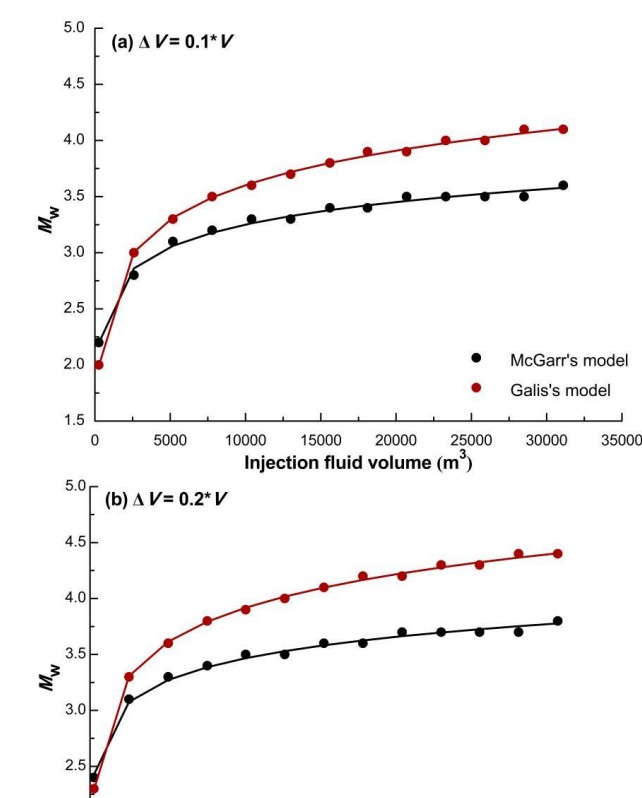



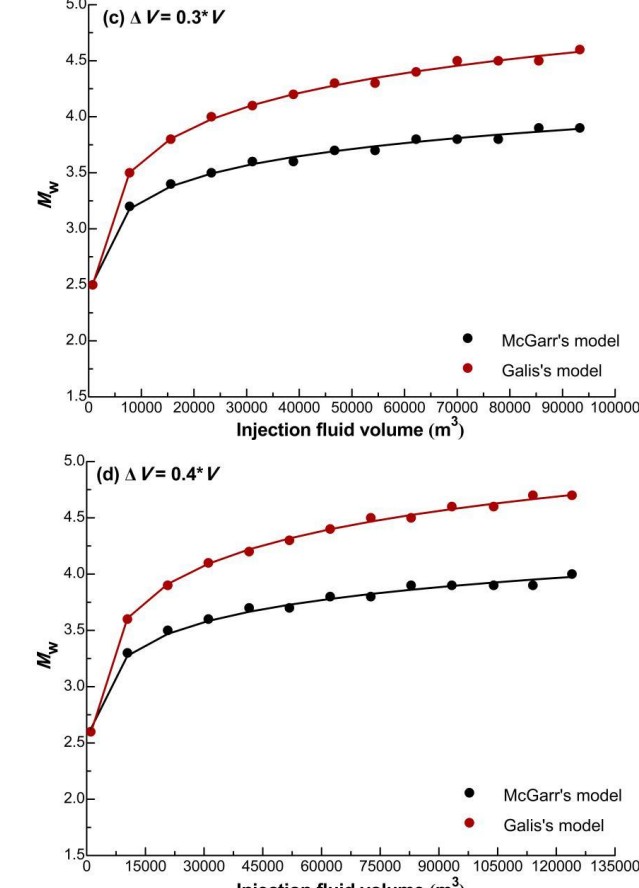

**Figure 19.** The maximum moment magnitude of injection-induced seismicity within ~10-15 km away from the
injected wells in response to monthly time series of injection volume in MTY EGS field. In the calculation, the
monthly injection volumes ($V$) are taken as $2.59\times10^3$ m³, $2.59\times10^4$ m³, $5.18\times10^4$ m³, $7.78\times10^4$ m³, $1.04\times10^5$ m³,
$1.30\times10^5$ m³, $1.56\times10^5$ m³, $1.81\times10^5$ m³, $2.07\times10^5$ m³, $2.33\times10^5$ m³, $2.59\times10^5$ m³, $2.85\times10^5$ m³ and $3.11\times10^5$ m³,
respectively, corresponding to fluid injected rate of 1 L/s, 10 L/s, 20 L/s, 30 L/s, 40 L/s, 50 L/s, 60 L/s, 70 L/s, 80
L/s, 90 L/s, 100 L/s, 110 L/s and 120 L/s, respectively.

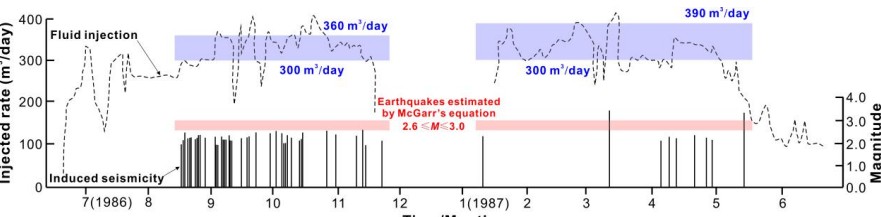

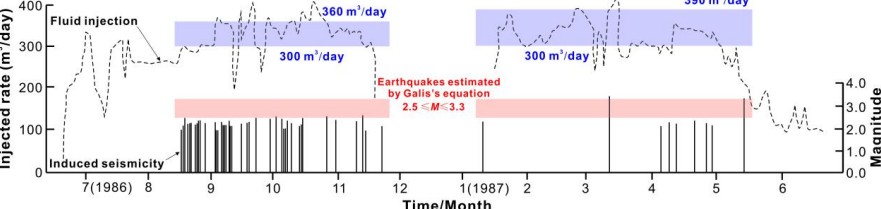

**Figure 20.** Comparisons of the magnitudes of the injection-induced earthquakes between the observations (black

vertical lines) (Zhao and Yang, 1990) and the predicted (pink rectangle) results calculated using the McGarr

model **(a)** and the Galis model **(b)**, in the Renqiu oilfield of North China.





**Table 1.** General conditions of three in situ stress boreholes measured in Tangshan seismic region.

| Name | Latitude,N | Longitude,E | Altitude,m | Depth,m | Rock type and integrity |
|------|-----------|-------------|-----------|---------|-------------------------|
| QABH | 40.12° | 118.81° | 38 | 600 | Being fine grained quartzite of the Archean and relatively intact. |
| CLBH | 39.74° | 119.15° | 56 | 600 | Being medium-coarse grained adamellite of the late Yanshanian Stage and well intact. |
| LXBH | 39.62° | 118.78° | 39 | 1000 | Being biotite leptynite and locally intercalated with some magnetite quartzite of the Archean, and relatively intact. |

**Table 2.** Hydraulic fracturing in situ stress results at QABH, CLBH, and LXBH boreholes near the MTY EGS
field.

| Name | Depth, m | Hydraulic fracturing parameter, MPa | | | | | Principal stress, MPa | | | Azimuth of $\sigma_H$ |
|------|----------|------|------|------|------|------|------|------|------|------|
| | | $P_0$ | $P_b$ | $P_r$ | $P_s$ | $T$ | $\sigma_H$ | $\sigma_h$ | $\sigma_v$ | |
| QABH | 184.10 | 1.67 | 15.36 | 11.53 | 8.31 | 3.83 | 15.08 | 8.31 | 4.88 | N47°W |
| | 208.90 | 1.92 | 15.16 | 10.31 | 7.40 | 4.85 | 13.82 | 7.40 | 5.54 | / |
| | 212.92 | 1.96 | 11.57 | 7.18 | 6.39 | 4.39 | 12.43 | 5.66 | 5.64 | N82°W |
| | 225.68 | 2.09 | 12.09 | 8.21 | 7.77 | 3.88 | 17.19 | 7.77 | 5.98 | / |
| | 251.65 | 2.35 | 17.16 | 12.49 | 9.69 | 4.67 | 18.93 | 9.69 | 6.67 | N57°W |
| | 305.88 | 2.89 | 12.67 | 7.03 | 6.83 | 5.64 | 16.36 | 6.83 | 8.11 | / |
| | 335.28 | 3.19 | 14.92 | 10.23 | 9.17 | 4.69 | 20.46 | 9.17 | 8.88 | / |
| | 380.68 | 3.64 | 12.71 | 8.30 | 7.12 | 4.41 | 16.70 | 7.12 | 10.09 | / |
| | 468.68 | 4.52 | 16.85 | 13.41 | 10.44 | 3.44 | 22.43 | 10.44 | 12.42 | N78°W |
| | 475.03 | 4.58 | 17.93 | 14.78 | 12.66 | 3.15 | 27.79 | 12.66 | 12.59 | / |
| | 511.30 | 4.95 | 19.68 | 14.71 | 12.76 | 4.97 | 28.51 | 12.76 | 13.55 | / |
| | 543.50 | 5.27 | 16.69 | 12.31 | 11.13 | 4.38 | 26.34 | 11.13 | 14.40 | / |
| | 562.00 | 5.45 | 15.84 | 10.41 | 9.61 | 5.43 | 23.88 | 9.61 | 14.89 | N62°W |
| | 590.32 | 5.74 | 16.08 | 12.18 | 10.68 | 3.90 | 25.59 | 10.68 | 15.64 | / |
| CLBH | 79.50 | 0.70 | 12.25 | 7.43 | 4.63 | 4.82 | 5.76 | 4.63 | 2.11 | N60°E |
| | 94.50 | 0.85 | 13.41 | 7.36 | 5.62 | 6.05 | 8.65 | 5.62 | 2.50 | / |
| | 119.50 | 1.10 | 19.78 | 6.11 | 3.75 | 13.67 | 4.04 | 3.75 | 3.17 | / |
| | 137.50 | 1.28 | 16.76 | 6.58 | 4.14 | 10.18 | 4.56 | 4.14 | 3.64 | / |
| | 206.50 | 1.97 | 18.24 | 6.00 | 5.20 | 12.24 | 7.63 | 5.20 | 5.47 | N84°E |
| | 248.10 | 2.38 | 16.94 | 6.62 | 5.56 | 10.32 | 7.68 | 5.56 | 6.57 | / |
| | 255.50 | 2.46 | 13.59 | 5.64 | 5.17 | 7.95 | 7.41 | 5.17 | 6.77 | / |
| | 290.50 | 2.81 | 14.02 | 7.71 | 6.18 | 6.31 | 8.02 | 6.18 | 7.70 | N82°W |
| | 335.50 | 3.26 | 16.03 | 8.46 | 7.75 | 7.57 | 11.53 | 7.75 | 8.89 | / |
| | 365.50 | 3.56 | 18.16 | 9.23 | 7.34 | 8.93 | 9.23 | 7.34 | 9.69 | N79°E |




| | | | | | | | | | |
|---|---|---|---|---|---|---|---|---|---|
| | 390.20 | 3.80 | 10.41 | 7.94 | 7.13 | 2.47 | 9.65 | 7.13 | 10.34 | / |
| | 393.50 | 3.94 | 14.26 | 9.75 | 9.18 | 4.51 | 13.95 | 9.18 | 10.43 | / |
| | 476.30 | 4.66 | 19.34 | 13.92 | 9.87 | 5.42 | 11.03 | 9.87 | 12.62 | / |
| | 485.50 | 4.76 | 24.65 | 13.61 | 10.62 | 11.04 | 13.49 | 10.62 | 12.87 | N55°E |
| LXBH | 340.00 | 3.36 | 21.50 | 16.26 | 11.53 | 5.24 | 14.97 | 11.53 | 9.01 | N60°W |
| | 386.00 | 3.82 | 24.42 | 18.82 | 14.38 | 5.60 | 20.50 | 14.38 | 10.23 | / |
| | 413.50 | 4.10 | 30.56 | 17.59 | 14.22 | 12.97 | 20.97 | 14.22 | 10.96 | N70°W |
| | 450.78 | 4.47 | 28.07 | 16.65 | 13.38 | 11.42 | 19.02 | 13.38 | 11.95 | / |
| | 506.00 | 5.02 | 28.19 | 16.89 | 13.49 | 11.30 | 18.56 | 13.49 | 13.41 | / |
| | 557.00 | 5.53 | 25.46 | 17.55 | 14.37 | 7.91 | 20.03 | 14.37 | 14.76 | / |
| | 605.00 | 6.01 | 24.79 | 18.05 | 15.40 | 6.74 | 22.14 | 15.40 | 16.03 | N74°W |
| | 649.26 | 6.45 | 26.20 | 17.97 | 15.49 | 8.23 | 22.05 | 15.49 | 17.21 | / |
| | 700.00 | 6.96 | 28.83 | 19.61 | 17.29 | 9.22 | 25.30 | 17.29 | 18.55 | / |
| | 748.00 | 7.44 | 29.97 | 19.00 | 16.48 | 10.97 | 23.00 | 16.48 | 19.82 | / |
| | 818.00 | 8.14 | 27.71 | 19.76 | 17.04 | 7.95 | 23.22 | 17.04 | 21.68 | / |
| | 922.44 | 9.18 | 27.70 | 20.96 | 19.46 | 6.84 | 28.24 | 19.46 | 24.44 | / |

Being noted that: $P_b$, $P_r$, and $P_s$ denote the breakdown, reopening and shut-in pressure, respectively. $\sigma_H$, $\sigma_h$, and $\sigma_v$
denote the maximum, minimum horizontal and vertical principal stresses, respectively. $P_0$ is the natural pore
pressure, and $T$ indicates the tensile strength of rock being equal to the difference between $P_b$ and $P_r$.
**Table 3.** The tensorial mean of hydraulic fracturing in situ stress results at similar depths in three boreholes near
the MTY EGS field.

| Name | Depth, m | In situ stress measured results, MPa | | | Average azimuth | In situ stress tensors, MPa | | | Tensorial mean of in situ stress, MPa | | | Estimated azimuth of $\sigma_H$ |
|---|---|---|---|---|---|---|---|---|---|---|---|---|
| | | $\sigma_H$ | $\sigma_h$ | Azimuth of $\sigma_H$ | | $\sigma_{xx}$ | $\sigma_{yy}$ | $\sigma_{xy}$ | $\sigma^a_{xx}$ | $\sigma^a_{yy}$ | $\sigma^a_{xy}$ | |
| QABH | 212.92 | 12.43 | 5.66 | 98° | 91° | 12.30 | 5.79 | 0.93 | 9.95 | 5.51 | 0.34 | 86° |
| CLBH | 206.50 | 7.63 | 5.20 | 84° | | 7.60 | 5.23 | -0.25 | | | | |
| LXBH | 340.00 | 14.97 | 11.53 | 120° | 99.5° | 14.11 | 12.39 | 1.49 | 11.64 | 9.90 | 0.57 | 73° |
| CLBH | 365.50 | 9.23 | 7.34 | 79° | | 9.16 | 7.41 | -0.35 | | | | |
| QABH | 468.68 | 22.43 | 10.44 | 102° | 78.5° | 21.91 | 10.96 | 2.44 | 17.23 | 11.26 | 0.54 | 85° |
| CLBH | 485.50 | 13.49 | 10.62 | 55° | | 12.55 | 11.56 | -1.35 | | | | |

Being noted that: Normal compressive stress is defined as positive, while tensional stress is negative. Shear stress
is positive (block clockwise rotation). For the *x-o-y* coordinate system, the positive direction of the *x*-axis and
*y*-axis are the same as the azimuth of East (E) and North (N), respectively.



3 **Table 4.** The geological information of the main active faults used for calculating probabilistic fault slip potential

4 in the Tangshan seismic region.

| Name | Segment | Length, km | strike | dip | Coefficient of friction |
|---|---|---|---|---|---|
| Jiyunhe fault ($F_2$) | $F_{2-1}$ | 25.37 | 292°±5° | 70°±10° | 0.4±0.05 |
| | $F_{2-2}$ | 29.37 | 315° ±5° | 70°±10° | 0.4±0.05 |
| | $F_{2-3}$ | 22.25 | 305°±5° | 70°±10° | 0.4±0.05 |
| Yejituo fault ($F_3$) | $F_{3-1}$ | 32.04 | 58°±5° | 60°±10° | 0.4±0.05 |
| | $F_{3-2}$ | 40.50 | 62°±5° | 60°±10° | 0.4±0.05 |
| | $F_{3-3}$ | 18.69 | 77°±5° | 60°±10° | 0.4±0.05 |
| | $F_{3-4}$ | 12.02 | 276°±5° | 60°±10° | 0.4±0.05 |
| Tangshan fault belt ($F_4$) | $F_{4-1}$ | 33.38 | 50°±5° | 80°±10° | 0.4±0.05 |
| | $F_{4-2}$ | 18.47 | 39°±5° | 80°±10° | 0.4±0.05 |
| | $F_{4-3}$ | 12.02 | 53°±5° | 80°±10° | 0.4±0.05 |
| | $F_{4-4}$ | 11.57 | 74°±5° | 80°±10° | 0.4±0.05 |
| | $F_{4-5}$ | 11.13 | 60°±5° | 80°±10° | 0.4±0.05 |
| | $F_{4-6}$ | 13.80 | 50°±5° | 80°±10° | 0.4±0.05 |
| Changli-Ninghe fault ($F_5$) | $F_{5-1}$ | 13.80 | 62°±5° | 75°±10° | 0.4±0.05 |
| | $F_{5-2}$ | 17.80 | 75°±5° | 75°±10° | 0.4±0.05 |
| | $F_{5-3}$ | 13.57 | 60°±5° | 75°±10° | 0.4±0.05 |
| | $F_{5-4}$ | 13.13 | 54°±5° | 75°±10° | 0.4±0.05 |
| | $F_{5-5}$ | 36.05 | 73°±5° | 75°±10° | 0.4±0.05 |
| | $F_{5-6}$ | 18.47 | 68°±5° | 75°±10° | 0.4±0.05 |
| | $F_{5-7}$ | 34.71 | 62°±5° | 75°±10° | 0.4±0.05 |
| | $F_{5-8}$ | 12.91 | 60°±5° | 75°±10° | 0.4±0.05 |
| | $F_{5-9}$ | 27.59 | 65°±5° | 75°±10° | 0.4±0.05 |
| Lulong fault ($F_6$) | $F_{6-1}$ | 22.92 | 22°±5° | 75°±10° | 0.4±0.05 |
| | $F_{6-2}$ | 19.58 | 27°±5° | 75°±10° | 0.4±0.05 |
| Luanxian-Laoting fault ($F_7$) | $F_{7-1}$ | 18.25 | 350°±5° | 70°±10° | 0.4±0.05 |
| | $F_{7-2}$ | 6.68 | 325°±5° | 70°±10° | 0.4±0.05 |
| | $F_{7-3}$ | 20.69 | 305°±5° | 70°±10° | 0.4±0.05 |
| | $F_{7-4}$ | 24.03 | 310°±5° | 70°±10° | 0.4±0.05 |
| | $F_{7-5}$ | 17.80 | 303°±5° | 70°±10° | 0.4±0.05 |
| Baigezhuang fault ($F_8$) | $F_{8-1}$ | 22.70 | 317°±5° | 65°±10° | 0.4±0.05 |
| | $F_{8-2}$ | 8.46 | 316°±5° | 65°±10° | 0.4±0.05 |
| | $F_{8-3}$ | 29.37 | 307°±5° | 65°±10° | 0.4±0.05 |
| | $F_{8-4}$ | 18.25 | 300°±5° | 65°±10° | 0.4±0.05 |
| Qinbei fault ($F_9$) | $F_{9-1}$ | 41.39 | 72°±5° | 60°±10° | 0.4±0.05 |
| | $F_{9-2}$ | 17.80 | 60°±5° | 60°±10° | 0.4±0.05 |
| | $F_{9-3}$ | 9.79 | 56°±5° | 60°±10° | 0.4±0.05 |
| Xi'nanzhuang fault ($F_{11}$) | $F_{11-1}$ | 20.03 | 86°±5° | 80°±10° | 0.4±0.05 |
| | $F_{11-2}$ | 13.35 | 64°±5° | 80°±10° | 0.4±0.05 |
| | $F_{11-3}$ | 13.35 | 70°±5° | 80°±10° | 0.4±0.05 |


| Tectonic units | Segment | Length/km | strike | dip | Coefficient of the friction |
|---|---|---|---|---|---|
| Haihe fault ($F_{12}$) | $F_{12\text{-}1}$ | 18.25 | 90°±5° | 70°±10° | 0.4±0.05 |
| | $F_{12\text{-}2}$ | 12.46 | 280°±5° | 70°±10° | 0.4±0.05 |
| | $F_{12\text{-}3}$ | 11.13 | 90°±5° | 70°±10° | 0.4±0.05 |
| | $F_{12\text{-}4}$ | 33.82 | 285°±5° | 70°±10° | 0.4±0.05 |
| Hangu fault ($F_{13}$) | $F_{13\text{-}1}$ | 15.58 | 288°±5° | 82°±10° | 0.4±0.05 |
| | $F_{13\text{-}2}$ | 10.68 | 90°±5° | 82°±10° | 0.4±0.05 |
| | $F_{13\text{-}3}$ | 11.57 | 285°±5° | 82°±10° | 0.4±0.05 |
| Cangdong fault ($F_{14}$) | $F_{14\text{-}1}$ | 25.81 | 43°±5° | 48°±10° | 0.4±0.05 |
| | $F_{14\text{-}2}$ | 9.79 | 60°±5° | 48°±10° | 0.4±0.05 |
| | $F_{14\text{-}3}$ | 21.81 | 77°±5° | 48°±10° | 0.4±0.05 |
| Lengkou fault ($F_{17}$) | $F_{17\text{-}1}$ | 34.27 | 291°±5° | 65°±10° | 0.4±0.05 |
| | $F_{17\text{-}2}$ | 39.16 | 286°±5° | 65°±10° | 0.4±0.05 |
| | $F_{17\text{-}3}$ | 20.03 | 292°±5° | 65°±10° | 0.4±0.05 |
| | $F_{17\text{-}4}$ | 14.24 | 320°±5° | 65°±10° | 0.4±0.05 |

**Table 5.** The geological information of the main boundary faults used for calculating probabilistic fault slip
potential in the MTY EGS field.

| Tectonic units | Segment | Length/km | strike | dip | Coefficient of the friction |
|---|---|---|---|---|---|
| Between II and I | $F_{b1}$ | 3.85 | 35°±5° | 45°±10° | 0.4±0.05 |
| | $F_{b2}$ | 3.21 | 20°±5° | 45°±10° | 0.4±0.05 |
| | $F_{b3}$ | 5.64 | 10°±5° | 45°±10° | 0.4±0.05 |
| | $F_{b4}$ | 1.28 | 50°±5° | 45°±10° | 0.4±0.05 |
| | $F_{b5}$ | 4.36 | 100°±5° | 45°±10° | 0.4±0.05 |
| | $F_{b6}$ | 7.56 | 95°±5° | 45°±10° | 0.4±0.05 |
| Between II and III | $F_{b7}$ | 16.67 | 110°±5° | 80°±10° | 0.4±0.05 |
| Between III and IV | $F_{b8}$ | 3.08 | 80°±5° | 80°±10° | 0.4±0.05 |
| | $F_{b9}$ | 3.33 | 60°±5° | 80°±10° | 0.4±0.05 |
| | $F_{b10}$ | 4.23 | 76°±5° | 80°±10° | 0.4±0.05 |
| Between II and IV | $F_{b11}$ | 6.15 | 50°±5° | 80°±10° | 0.4±0.05 |
| | $F_{b12}$ | 3.72 | 63°±5° | 80°±10° | 0.4±0.05 |
| | $F_{b13}$ | 8.21 | 25°±5° | 80°±10° | 0.4±0.05 |
| | $F_{b14}$ | 5.77 | 55°±5° | 80°±10° | 0.4±0.05 |
| Southwestern II | $F_{b15}$ | 4.87 | 130°±5° | 65°±10° | 0.4±0.05 |
| Between IV and V | $F_{b16}$ | 3.85 | 84°±5° | 80°±10° | 0.4±0.05 |
| | $F_{b17}$ | 2.69 | 70°±5° | 80°±10° | 0.4±0.05 |
| | $F_{b18}$ | 9.10 | 90°±5° | 80°±10° | 0.4±0.05 |
| | $F_{b19}$ | 3.85 | 120°±5° | 80°±10° | 0.4±0.05 |
| | $F_{b20}$ | 3.08 | 97°±5° | 80°±10° | 0.4±0.05 |

