# Peer review of "Fault slip potential induced by fluid injection in the"

_Natural Hazards and Earth System Sciences, 2022_

## Referee Comment (RC1)

**Review Comments**

This paper provides a thorough investigation of the fault slip potential induced by fluid injection in the Matouying EGS field, Tangshan, China. The case study of the EGS field has important implications for deep geothermal exploitations. Overall, the paper is interesting, useful, and well-written. I believe this paper should be considered for publication if the authors could address the following comments/suggestions.

1. In Section "**4.1 Stress field inversion from earthquake focal mechanisms**" on page 5, it would be better if the authors could provide the focal mechanisms data in the supplement.

2. In Section "**5 Initial fault slip potential in Tangshan seismic region in the present stress field**" on pages 8 and 9, if the deterministic geomechanical assessment ignores multiple sources of uncertainty, and if the probabilistic geomechanical assessment is more robust and accurate, is it important to also present the deterministic results? If the deterministic assessment provides compelling or useful information, consider adding more information into the paper as to why this information is relevant. Otherwise, consider removing the deterministic analysis from this paper.

3. In Section "**6.1 Hydrology model**" on page 12, whenever you mention the simplifying assumptions of a method, consider adding some commentary as to whether these assumptions are appropriate or not in the MYT EGS field? Please clarify.

4. In Section "**8 Discussion**", I suggest that the authors should also discuss the effect of porosity on the fault slip potential in the MTY EGS field.

5. In Section "**8.4 The predicted maximum magnitude of injection-induced seismicity in MTY EGS field**", what is your explanation for the discrepancies in the Galis and McGarr model results? What are the strengths and weaknesses of these two models?

6. In Section "**8.4 The predicted maximum magnitude of injection-induced seismicity in MTY EGS field**", whether the predicted maximum magnitude of

injection-induced seismicity would be larger or smaller than that of the largest natural earthquake with a magnitude in the MTY EGS field? Please make some comparisons.

7. In Figure 11, Fig.11(b) should be the probabilistic fault slip potential on the mapped faults in response to the hypothetic fluid injection in 2040, rather than 2030.

8. Regarding the paper organization, there are too many sections, and some of them could be combined. For example, Sections 2 and 3 could be combined as a background section; Sections 5, 6 and 7 could also be combined.

---

## Author Comment (AC4)

Dear Reviewer,

Thanks for reviewing our manuscript. We all agree with you completely. We will revise our manuscript based on your helpful comments after interactive discussion.

1) The focal mechanisms dataset used for the inversion should be provided as a table in the supplementary material.

**Response 01: We will provide the focal mechanisms dataset used for inversion in the supplementary materal in our revised manuscript.**

2) There are too many sections and figures in the manuscript. Some of the sections can be combined, as for instance sections 3, 5.1, 6.1 combined in a methodology section and sections 5, 6 and 7 in a broader results section. Some of the figures can be transferred to the supplementary material, or provided as insets in other figures (e.g., Fig.9 as inset of Fig.8).

**Response 02: We will combine some sections together, and some of the figures will be transferred to the supplementary material in our revised manuscript later.**

3) In the local scale of Fig.13, the authors display several faults in the vicinity of the MTY EGS that are not displayed in the regional maps (e.g., Fig.11). Discuss the reason and perhaps provide local scale figures as insets in the regional maps to reduce the number of figures.

**Response 03: Due to different graphic scale of Fig.13 and Fig. 11 (et al), several faults in the vicinity of the MTY EGS are not displayed in the regional maps. We will provide local scale figures as insets in the regional maps in our revised manuscript later.**

Some minor comments related to the text concern:

1) Page 1, Line 34: define the abbreviation "FSP".

2) Provide the definition of μ in Eq.(1).

3) In Page 9, Line 32, the numbering of the referred equations is probably wrong.

4) Replace 2030 with 2040 in Fig.11b or replace with the correct figure.

5) Correct the first word in Page 16, Line 1.

**Response 04: We will revise these minor errors in our revised manuscript later.**

---

## Author Comment (AC5)

Dear Reviewer01,

Thanks for reviewing our manuscript. We all agree with you completely. We will revise our manuscript based on your helpful comments after interactive discussion.

1. In Section "4.1 Stress field inversion from earthquake focal mechanisms" on page 5, it would be better if the authors could provide the focal mechanisms data in the supplement.

**Response 01: We will provide the focal mechanisms dataset used for inversion in the supplementary materal in our revised manuscript.**

2. In Section "5 Initial fault slip potential in Tangshan seismic region in the present stress field" on pages 8 and 9, if the deterministic geomechanical assessment ignores multiple sources of uncertainty, and if the probabilistic geomechanical assessment is more robust and accurate, is it important to also present the deterministic results? If the deterministic assessment provides compelling or useful information, consider adding more information into the paper as to why this information is relevant. Otherwise, consider removing the deterministic analysis from this paper.

**Response 02: The deterministic assessment should be still necessary. Before we estimate the fault slip potential on mapped faults in the Tangshan seismic region, the deterministic assessment may provide deterministic fluid pressure to slip. Then, we will comprehensively know much about the initial stability of these faults, and mainly discuss the effects of fluid injection on these faults that may have higher fault slip potential in response to smaller fluid pressure perturbations.**

3. In Section "6.1 Hydrology model" on page 12, whenever you mention the simplifying assumptions of a method, consider adding some commentary as to

whether these assumptions are appropriate or not in the MYT EGS field? Please clarify.

**Response 03: The MTY EGS field lies in the gneiss that are well compact and intact with the porosity of less than 6%, and there are fewer pre-existing fractures at the depth interval of 3965-4000 m. Therefore, the Hsieh and Bredehoeft's hydrology model should be appropriate for the MTY EGS field.**

4. In Section "8 Discussion", I suggest that the authors should also discuss the effect of porosity on the fault slip potential in the MTY EGS field.

**Response 04: We have published a paper titled "Changes in Fault Slip Potential Due to Water Injection in the Rongcheng Deep Geothermal Reservoir, Xiong'an New Area, North China" (Water, 2022,14,410, https://doi.org/10.3390/w14030410). In this paper, we discuss the effect of porosity o the fault slip potential (FSP) values, and our results showed that the FSP values do not have obvious changes with increasing porosity. However, in this manuscript, we will calculate the effect of porosity on the FSP values on the mapped faults in the MTY EGS field again in our revised manuscript later.**

5. In Section "8.4 The predicted maximum magnitude of injection-induced seismicity in MTY EGS field", what is your explanation for the discrepancies in the Galis and McGarr model results? What are the strengths and weaknesses of these two models?

**Response 05: In our manuscript, by comparison, we find that the maximum magnitudes of the injection-related seismicity estimated with the Galis model are slightly greater than the values by the McGarr model. We find that the maximum magnitudes of the injection-related seismicity estimated with the Galis model are more similar with the observed earthquked in the Renqiu oil field, North China. Besides, we will consider**

**(and discuss) also other commonly accepted models to estimate the expected maximum magnitude (e.g., Shapiro et al. (2011), Van der Elst et al. (2016)) in our revised manuscript later.**

6. In Section "8.4 The predicted maximum magnitude of injection-induced seismicity in MTY EGS field", whether the predicted maximum magnitude of injection-induced seismicity would be larger or smaller than that of the largest natural earthquake with a magnitude in the MTY EGS field? Please make some comparisons.

**Response 06: As shown in Fig. 19d, when the accumulated net injected volume is larger than 5000 L and the 5 fluid loss is 40%, the predicted Mw slowly increases from $M_w$ 3.3 to $M_w$ 4.0 with the McGarr model and from $M_w$ 3.6 to $M_w$ 4.7 with the Galis model. For 40% fluid loss, the maximum $M_w$ of an injection-induced earthquake in the MTY EGS field is $M_w$ 4.7, that is more smaller than the natural earthquake in the Tangshan seismic region (e.g., 1976.7.28, M 7.8). We suggest that the maximum magnitudes of the injection-related seismicity in the MTY EGS could be no more than $M_w$ 6.0.**

7. In Figure 11, Fig.11(b) should be the probabilistic fault slip potential on the mapped faults in response to the hypothetic fluid injection in 2040, rather than 2030.

**Response 07:We will modify this error in our revised manuscript later.**

8. Regarding the paper organization, there are too many sections, and some of them could be combined. For example, Sections 2 and 3 could be combined as a background section; Sections 5, 6 and 7 could also be combined.

**Response 08:We will combine some sections together in our revised manuscript later.**

---

## Author Comment (AC6)

Dear Reviewer03,

Thanks for reviewing our manuscript. We will revise our manuscript based on your helpful comments after interactive discussion.

1) I suggest to use the thermoporoelasticity equation instead of the simple elasticity equation (eq. 1 in your manuscript) to evaluate the critical shear stress. This will help you to consider also the effect of temperature changes on stress, in addition to your different injection scenarios, in a more comprehensive way with respect to what you did in section 8.3.

**Response 01:** **We were not able to fully comprehend parts of the this comment but strived to provide our response based on our best interpretation.**

**(1) In our manuscript, Eq.1 presents a useful framework for understanding how increasing the pore fluid pressure via fluid injection can trigger fault slip under different in-situ stress conditions.**

**(2) In Eq.1, the $\sigma_n$ and $\tau_c$ can consider all stress effects, including elasticity, poroelasticity, and porothermoelasticity, et al.**

**(3) In our manuscript, we first calculate the pore pressure induced by fluid injection and discuss the fault slip potential associated with these pore pressure perturbations.**

**(4) In section 8.3, we also calculate the thermoelasticity based on thermally uncoupled solution (Cheng, 2016, Poroelasticity,Theory and Applications of Transport in Porous Media, Springer Nature ), and discuss the effect of thermoelasticity on the fault slip potential on the mapped faults near the MTY EGS field.**

**(5) Our results suggest that the stress changes (1.25 MPa) with changes in temperature (by 6 °C during 20 years) have less influence on the faulting slip trendency than the direct pore pressure perturbations induced by fluid injection. Therefore, in our manuscript, we mainly**

**discuss the effect of pore pressure perturbations on the faulting stability in the MTY EGS field.**

**(6) Presently, preliminary hydraulic stimulation tests have been conducted with smaller injection rate, and there have been lack of observed induced earthquakes in the MTY EGS field. In the future, we will focus on the poroelasticity on the induced seismicity in the MTY EGS production at different depths.**

2) It would be interesting if you will consider (and discuss) also other commonly accepted models to estimate the expected maximum magnitude (e.g., Shapiro et al. (2011), Van der Elst et al. (2016)).

**Response 02: In our manuscript, by comparison, we find that the maximum magnitudes of the injection-related seismicity estimated with the Galis model are slightly greater than the values by the McGarr model. Besides, we will consider (and discuss) also other commonly accepted models to estimate the expected maximum magnitude (e.g., Shapiro et al. (2011), Van der Elst et al. (2016)) in our revised manuscript later.**

Additional minor comment:

- At line 19 of the Abstract I suggest to substitute "Enhanced Geothermal Systems (EGS) field" with "Enhanced Geothermal System (EGS) field"

**Response 03: We will modify this minor error in our revised manuscript later.**

---

## Author Response (AR1)

**Dear editor and reviewers,**

Thanks a lot for reviewing our manuscript. We have substantially revised our manuscript after reading the comments provided by the editor and three reviewers. We again thank all of you for considering our work and look forward to your response at your earliest convenience. Please direct all correspondence about this manuscript to Dr.Chengjun Feng (corresponding author in the current version, Institute of Geomechanics, Chinese Academy of Geological Sciences).

Sincerely

Dr.Chengjun Feng (feng2010618@aliyun.com)

17, May, 2022, Beijing, China.

**Answers to reviewers by authors:**

--RC #1:

**1. In Section "4.1 Stress field inversion from earthquake focal mechanisms" on page 5, it would be better if the authors could provide the focal mechanisms data in the supplement.**

**Answer:** We provided the focal mechanisms dataset used for inversion in the supplementary materal (Table S1) in our revised manuscript (see page 7, line 21).

**2. In Section "5 Initial fault slip potential in Tangshan seismic region in the present stress field" on pages 8 and 9, if the deterministic geomechanical assessment ignores multiple sources of uncertainty, and if the probabilistic geomechanical assessment is more robust and accurate, is it important to also present the deterministic results? If the deterministic assessment provides compelling or useful information, consider adding more information into the paper as to why this information is relevant. Otherwise, consider removing the deterministic analysis from this paper.**

**Answer:** The deterministic assessment should be still necessary. Before we estimate the fault slip potential on mapped faults in the Tangshan seismic region, the

deterministic assessment may provide deterministic fluid pressure to slip. Then, we will comprehensively know much about the initial stability of these faults, and mainly discuss the effects of fluid injection on these faults that may have higher fault slip potential in response to smaller fluid pressure perturbations. Therefore, in our revised manuscript, we still calculate both deterministic and probabilistic fault slip potential in Tangshan seismic region under the present stress field (see sections 5.1 and 5.2 from page 10 to 12).

**3. In Section "6.1 Hydrology model" on page 12, whenever you mention the simplifying assumptions of a method, consider adding some commentary as to whether these assumptions are appropriate or not in the MYT EGS field? Please clarify.**

**Answer:** The MTY EGS field lies in the gneiss unit of the Bai'miaozi Series in the Dan'tazi Group of the Archaeozoic Erathem, and has an area of 80 km$^2$ at the depth of 4000 m (Qi et al., 2020; Zhang et al., 2020). The thickness of preliminary hydraulic stimulation test is 35 m (depth range of 3965－4000 m) (Wang et al., 2013; Zhang et al., 2014). As a further simplication, horizontal two-dimensional flow is assumed. In additon, at the depth interval of 3965-4000 m, the gneiss are well compact and intact with fewer pre-existing fractures (Zhang et al., 2022). In these respects, the MTY EGS unit can be taken as an infinite and isotropic reservoir. Thus, the Hsieh and Bredehoeft's hydrology model (Eqs. (2) and (3) in our revised manuscript) should be acceptable for the MTY EGS field. The details are also shown in page 12 from line 9 to 16.

**4. In Section "8 Discussion", I suggest that the authors should also discuss the effect of porosity on the fault slip potential in the MTY EGS field.**

**Answer:** To further understand how porosity may influence the probabilistic FSP on the mapped faults near the MTY EGS field, we recaculated the FSP values for the $F_{b11}$, $F_{b12}$, $F_{b13}$, and $F_{b14}$ fault segments in 2030, 2040, and 2050, respectively, with porosity values ranging from 4.2% to 9.6% in 0.9% increment. Our results show that

the fault slip potential on the mapped faults induced by long-term fluid injection in the MTY EGS field changes slightly in response to variations in porosity. This finding is well consistent with our previous results in the Rongcheng geothermal reservoir of Xiong'an New Area, North China (Zhu et al., 2022). The details are also shown in our revised manuscript (see section 6.3 in pages 15 and 16).

**5. In Section "8.4 The predicted maximum magnitude of injection-induced seismicity in MTY EGS field", what is your explanation for the discrepancies in the Galis and McGarr model results? What are the strengths and weaknesses of these two models?**

**Answer:** For a fluid loss of 40%, when the accumulated net injected volume is larger than 5000 $m^3$, the predicted maximum $M_w$ of injection-induced earthuqakes is 4.0 with the McGarr model, and is 4.7 with the Galis model. In addition, we find that the maximum magnitudes of the injection-related seismicity estimated with the Galis model ($M_w$ 3.3) are slightly greater than the values by the McGarr model ($M_w$ 3.0) in Renqiu oil field, North China. By comparison, we also find that the maximum magnitudes of the injection-related seismicity estimated with the Galis model ($M_w$ 3.3) are more similar with the observed earthquked ($M_w$ 3.5) in the Renqiu oil field. In general, the defference in the expected maximum magnitudes estimated with these two models is less than 0.7. Based on this analysis, we conclude that these two of the predicted models can be adopted to estimate the maximum moment magnitude of fluid-injected seismic events in the MTY EGS field in anticipation of the upcoming EGS exploitation. The details are shown in section 6.5 from page 16 to 19.

**6. In Section "8.4 The predicted maximum magnitude of injection-induced seismicity in MTY EGS field", whether the predicted maximum magnitude of injection-induced seismicity would be larger or smaller than that of the largest natural earthquake with a magnitude in the MTY EGS field? Please make some comparisons.**

**Answer:** As shown in Fig.17 in our revised manuscript, we find that the

predicted maximum magnitude of earthquakes ($M_\text{w}$ 4.7) induced by continuous water injection for 30 days would be greatly smaller than that of the largest tectonic earthquake in Tangshan seismic region (i.e., the 1976 $M$ 7.8 earthquake). This finding is also shown in page 20 of Chapter 7 in our revised manuscript.

**7. In Figure 11, Fig.11(b) should be the probabilistic fault slip potential on the mapped faults in response to the hypothetic fluid injection in 2040, rather than 2030.**

**Answer:** We have modified this error in our revised manuscript (see Fig.10b in page 37).

**8. Regarding the paper organization, there are too many sections, and some of them could be combined. For example, Sections 2 and 3 could be combined as a background section; Sections 5, 6 and 7 could also be combined.**

**Answer:** In our revised manuscript, we combined sections 5.1, 5.2, 5.3, 5.4, and 5.5 together, as a new Chapter of of "5 Results".

**--RC #2:**

**1. The focal mechanisms dataset used for the inversion should be provided as a table in the supplementary material.**

**Answer:** We provided the focal mechanisms dataset used for inversion in the supplementary materal (Table S1) in our revised manuscript (see page 7, line 21).

**2. There are too many sections and figures in the manuscript. Some of the sections can be combined, as for instance sections 3, 5.1, 6.1 combined in a methodology section and sections 5, 6 and 7 in a broader results section. Some of the figures can be transferred to the supplementary material, or provided as insets in other figures (e.g., Fig.9 as inset of Fig.8).**

**Answer:** We combined sections 3, 5.1, and 6.1 of initial manuscript as a new Chapter "3 Methodology" in our revised manuscript (see pages 5 and 6). In addition,

we also combined sections 5.1, 5.2, 5.3, 5.4, and 5.5 together, as a new Chapter of of "5 Results". Fig.9 in initial manuscript have been transferred to the supplementary material in the revised manuscript (see in Fig.S3). Fig.12 in initial manuscript have been transferred to the supplementary material in the revised manuscript (see in Fig.S6). Fig.15 in initial manuscript have also been transferred to the supplementary material in the revised manuscript (see in Fig.S7).

**3. In the local scale of Fig.13, the authors display several faults in the vicinity of the MTY EGS that are not displayed in the regional maps (e.g., Fig.11). Discuss the reason and perhaps provide local scale figures as insets in the regional maps to reduce the number of figures.**

**Answer:** Due to different graphic scale of Fig.13 (larger) and Fig. 11 (et al) (smaller) in initial manuscript, several faults in the vicinity of the MTY EGS are not displayed in the regional maps. In the revised manuscript, we intended to provide local scale figure (Fig.11) as insets in the regional maps, however, being limited by the size of these regional maps, we still decided to display the faults near the MTY EGS field separately (Fig.11), and to estimate the FSP values on these mapped faults in different section (Section 5.5 in pages 13 and 14).

**4. Some minor comments related to the text concern:**

1) Page 1, Line 34: define the abbreviation "FSP".

**Answer:** In the "Abstract", we defined the abbreviation "FSP" in the trevised manuscript (line 33).

2) Provide the definition of μ in Eq.(1).

**Answer:** We provided the difinition of $\mu$ in Eq.(1) in the revised manuscript (see in page 4, line 38).

3) In Page 9, Line 32, the numbering of the referred equations is probably wrong.

**Answer:** In the revised manuscript, we modify this error (see in page 10, line 15).

4) Replace 2030 with 2040 in Fig.11b or replace with the correct figure.

**Answer:** We have modified this error in our revised manuscript (see Fig.10b in page 37).

5) Correct the first word in Page 16, Line 1.

**Answer:** The word "seiongoingsmic" of initial manuscript in page 16, line 1, has been corrected with "ongoing seismic" in the revised manuscript (in page 15, line 34).

**--RC #3:**

**1. I suggest to use the thermoporoelasticity equation instead of the simple elasticity equation (eq. 1 in your manuscript) to evaluate the critical shear stress. This will help you to consider also the effect of temperature changes on stress, in addition to your different injection scenarios, in a more comprehensive way with respect to what you did in section 8.3.**

**Answer:** We were not able to fully comprehend parts of the this comment but strived to provide our response based on our best interpretation.

1) In our revised manuscript, Eq.1 presents a useful framework for understanding how increasing the pore fluid pressure via fluid injection can trigger fault slip under different in-situ stress conditions.

2) In Eq.1, the $\sigma_n$ and $\tau_c$ can consider all stress effects, including elasticity, poroelasticity, and porothermoelasticity, et al.

3) In our revised manuscript, we first calculate the pore pressure induced by fluid injection and discuss the fault slip potential associated with these pore pressure perturbations.

4) In section 6.4, we also calculate the thermoelasticity based on thermally uncoupled solution (Cheng, 2016, Poroelasticity,Theory and Applications of Transport in Porous Media, Springer Nature), and discuss the effect of thermoelasticity on the fault slip potential on the mapped faults near the MTY EGS field.

5) Our results suggest that the stress changes (1.25 MPa) with changes in temperature (by 6 °C during 20 years) have less influence on the faulting slip trendency than the direct pore pressure perturbations induced by fluid injection.

Therefore, in our revised manuscript, we mainly discuss the effect of pore pressure perturbations on the faulting stability in the MTY EGS field.

6) Presently, preliminary hydraulic stimulation tests have been conducted with smaller injection rate, and there have been lack of observed induced earthquakes in the MTY EGS field. In the future, we will focus on the poroelasticity on the induced seismicity in the MTY EGS production at different depths.

**2. It would be interesting if you will consider (and discuss) also other commonly accepted models to estimate the expected maximum magnitude (e.g., Shapiro et al. (2011), Van der Elst et al. (2016)).**

**Answer:** In our revised manuscript, we considered the Van der Els model (Eq.13 in section 6.5) and estimated the expected maximum magnitudes of injection-induced earthquakes in the MTY EGS field, using the Galis model, the McGarr model, and the Van der Els model (see in pages 17, 18, and 19). In general, the defference in the expected maximum magnitudes estimated with these three models is less than 0.5. Based on this analysis, we conclude that these three of the predicted models can be adopted to estimate the maximum moment magnitude of fluid-injected seismic events in the MTY EGS field in anticipation of the upcoming EGS exploitation.

**3. Additional minor comment:** At line 19 of the Abstract I suggest to substitute "Enhanced Geothermal Systems (EGS) field" with "Enhanced Geothermal System (EGS) field".

**Answer:** We have modified this minor error in our revised manuscript (see in page 1, line 18).